# A reference map of murine cardiac transcription factor chromatin occupancy identifies dynamic and conserved enhancers

Brynn N. Akerberg [1,12], Fei Gu[1,2,12], Nathan J. VanDusen [1], Xiaoran Zhang[1], Rui Dong[3], Kai Li[1], Bing Zhang[4], Bin Zhou [5], Isha Sethi[1], Qing Ma[1], Lauren Wasson [6], Tong Wen[7], Jinhua Liu[8], Kunzhe Dong[9], Frank L. Conlon[6], Jiliang Zhou [9], Guo-Cheng Yuan [3,10], Pingzhu Zhou[1] & William T. Pu [1,11]*

Mapping the chromatin occupancy of transcription factors (TFs) is a key step in deciphering developmental transcriptional programs. Here we use biotinylated knockin alleles of seven key cardiac TFs (GATA4, NKX2-5, MEF2A, MEF2C, SRF, TBX5, TEAD1) to sensitively and reproducibly map their genome-wide occupancy in the fetal and adult mouse heart. These maps show that TF occupancy is dynamic between developmental stages and that multiple TFs often collaboratively occupy the same chromatin region through indirect cooperativity. Multi-TF regions exhibit features of functional regulatory elements, including evolutionary conservation, chromatin accessibility, and activity in transcriptional enhancer assays. H3K27ac, a feature of many enhancers, incompletely overlaps multi-TF regions, and multi-TF regions lacking H3K27ac retain conservation and enhancer activity. TEAD1 is a core component of the cardiac transcriptional network, co-occupying cardiac regulatory regions and controlling cardiomyocyte-specific gene functions. Our study provides a resource for deciphering the cardiac transcriptional regulatory network and gaining insights into the molecular mechanisms governing heart development.

[1] Department of Cardiology, Boston Children's Hospital, 300 Longwood Avenue, Boston, MA 02115, USA. [2] Alibaba Cloud Intelligence Business Group, Alibaba Group, 311121 Hangzhou, China. [3] Department of Biostatistics and Computational Biology, Dana-Farber Cancer Institute, 450 Brookline Avenue, Boston, MA 02215, USA. [4] Xin Hua Hospital, Key Laboratory of Systems Biomedicine, Shanghai Center for Systems Biomedicine, Shanghai Jiao Tong University, 200240 Shanghai, China. [5] Institute of Biochemistry and Cell Biology, Shanghai Institutes for Biological Sciences, 200031 Shanghai, China. [6] Biology Department, University of North Carolina at Chapel Hill, 120 South Road, Chapel Hill, NC 27599, USA. [7] Department of Cardiology, The First Affiliated Hospital of Nanchang University, 330006 Nanchang, China. [8] Department of Respiratory Medicine, The First Affiliated Hospital of Nanchang University, 330006 Nanchang, China. [9] Department of Pharmacology & Toxicology, Medical College of Georgia, Augusta University, 1459 Laney Walker Boulevard, Augusta, GA 30912, USA. [10] Department of Biostatistics, Harvard T.H. Chan School of Public Health, 677 Huntington Avenue, Boston, MA 02115, USA. [11] Harvard Stem Cell Institute, Harvard University, 7 Divinity Avenue, Cambridge, MA 02138, USA. [12]These authors contributed equally: Brynn N. Akerberg, Fei Gu. *email: william.pu@enders.tch.harvard.edu

Distal chromatin regulatory elements direct transcriptional programs that govern organ development by recruiting sequence-specific DNA-binding transcription factors (TFs), which nucleate transcriptional regulatory complexes and deposit activating epigenetic marks, such as histone H3 acetylation on lysine 27 (H3K27ac)[1]. Deciphering these transcriptional programs requires identifying their components, including the participating TFs and the chromatin sites to which they bind[2]. Transcription factor occupancy has been mapped primarily through chromatin immunoprecipitation followed by next-generation sequencing (ChIP-seq). These maps have been established largely for cell lines and are limited by the sensitivity and specificity of antibody-mediated chromatin immunoprecipitation[3,4]. As a result of these limitations, the transcriptional programs that govern development and homeostasis of most organs remain incompletely elucidated.

The development of the heart is orchestrated by intricate transcriptional programs, so that mutations in TFs and epigenetic regulators are important causes of congenital heart disease[5]. Among the well known cardiac TFs are NKX2-5, TBX5, GATA4, MEF2A, MEF2C, and SRF[6–11]. Although TEAD1, formerly known as TEF-1, has recently gained interest as the nuclear target of Hippo-YAP signaling[12], it was one of the first TFs implicated in heart development[13]. Biochemical and functional interactions between these TFs indicate that they collaboratively regulate cardiac gene expression[5,14].

We previously showed that pull-down of biotinylated TFs and associated chromatin followed by next generation sequencing (bioChIP-seq) sensitively and reproducibly maps TF occupancy, both in cultured cells and in mouse tissues[3,4,15,16]. Here we extend this approach to map the genome-wide occupancy of seven TFs (GATA4, MEF2A, MEF2C, NKX2-5, SRF, TBX5, and TEAD1) in fetal and adult heart, providing a valuable resource for the study of cardiac transcriptional regulation. Our analyses of these data identify regions bound by multiple TFs and show that these regions are transcriptional regulatory elements that function both in the presence and absence of the activating histone mark H3K27ac.

## Results

**Dynamic transcription factor chromatin occupancy.** We generated seven mouse knock-in lines in which an epitope tag encoding FLAG and a biotin acceptor peptide (BIO) was fused to the C-terminus of key cardiac TFs (GATA4$^{fb}$, MEF2A$^{fb}$, MEF2C$^{fb}$, NKX2-5$^{fb}$, SRF$^{fb}$, TBX5$^{fb}$, and TEAD1$^{fb}$; Fig. 1a, Supplementary Fig. 1 and refs. [16–19]). The knockin alleles expressed epitope-tagged protein at levels comparable to the wild-type allele (Supplementary Fig. 1 and refs. [16–19]). The mice were viable and fertile as homozygotes, with the exception of *Mef2c$^{fb/fb}$* mice, which died perinatally with ventricular septal defects and aortic override (Supplementary Fig. 1d). In contrast, *Mef2c* null mice died by embryonic day 10 (E10) with two chambered hearts that failed to undergo normal looping[9], indicating that *Mef2c$^{fb}$* is hypomorphic but sufficient to support most aspects of fetal heart development. Heterozygous knockin alleles supported normal heart function (Supplementary Fig. 2 and refs. [17,18]).

Biotin ligase, expressed from the *Rosa26* locus[20], recognizes and biotinylates the BIO peptide. High affinity pull-down of the resulting biotinylated TFs onto immobilized streptavidin followed by massively parallel sequencing (bioChIP-seq) permitted highly sensitive and reproducible genome-wide mapping of chromatin occupancy under consistent conditions, without being vulnerable to the potential idiosyncrasies of antibodies used for chromatin immunoprecipitation[3,4,15] (Fig. 1a). We performed bioChIP-seq for the seven TFs from heterozygous fetal (E12.5) and adult (P42)

ventricular apexes, in biological duplicate (Supplementary Table 1). Despite numerous attempts, adult heart MEF2C bioChIP-seq was not successful, likely because of its relatively low expression in the adult heart, where MEF2A and MEF2D are the predominant isoforms (Supplementary Fig. 3 and refs. [21–24]). The bioChIP-seq biological duplicates were tightly correlated (Fig. 1b). Samples showed greater correlation between factors within the same stage than between the same factor at different stages (Fig. 1b). Consistent with this, each TF occupied markedly different genomic regions between fetal and adult stages (Jaccard similarity between stages = 34 ± 15% (mean ± s.d.); Fig. 1c; Supplementary Fig. 4).

Replicate data were combined by retaining reproducible peaks[25]. In total, we identified 247,799 reproducible TF-binding peaks across the 13 samples (35,400 ± 14,760 peaks per sample, mean ± s.d.; Fig. 1d and Supplementary Data 1). The bioChIP-seq data overlapped moderately well with published mouse ChIP-seq data from heart or cardiomyocyte-related cultured cells (Supplementary Fig. 5a, b), considering the biological and technical differences between samples, and TF occupancy of the *Nppa-Nppb* gene cluster (Supplementary Fig. 5c, d) was similar to previous reports[26,27]. Cardiac TFs predominantly occupied genomic regions distal (>2 kb) to transcription start sites (TSSs; Fig. 1d). Distal TF regions were evenly distributed between intergenic and intronic regions (Fig. 1d).

Gene ontology (GO) analysis showed that each TF was enriched for a different set of biological process terms. Many of the TF's GO terms changed between developmental stages (Fig. 1e and Supplementary Data 2). For example, fetal SRF regions were enriched for actin cytoskeleton, whereas adult SRF regions were linked to muscle cell/myofibrils and metabolism, consistent with our recent study[16]. TEAD1 was enriched for terms related to heart morphogenesis and ion/transport in the fetal heart and actin cytoskeleton and metabolism in the adult heart.

Each TF's bound regions were most highly enriched for its own DNA-binding motif (Fig. 1f and Supplementary Fig. 6a; >33% of top 1000 regions). The exception was fetal MEF2C, where the MEF2 motif was less highly enriched (9.5% of top 1000 regions) than the NKX2-5 and TEAD motifs (48% and 47% of top 1000 regions). In contrast, MEF2A regions were most enriched for the MEF2 motif (53% of top 1000 regions) and only slightly enriched for NKX2-5 and TEAD motifs, despite 99% identity between MEF2A and MEF2C DNA-binding domains. This analysis was independently supported by central enrichment analysis, which compares motif frequency at ChIP-seq peak summits compared to flanking regions[28]. Central enrichment analysis identified highly significant over-representation of each bioChIP'd TF's motif at its peak summit, again with the exception of fetal MEF2C (Supplementary Fig. 6b). MEF2A regions showed strong central enrichment for both the MEF2 and SRF motifs (Fig. 1g, left); in contrast, MEF2C regions had weak central enrichment for MEF2 and SRF motifs and strong central enrichment for NKX2-5 and TEAD motifs (Fig. 1g, right).

Each TF's bound regions were also enriched for motifs of other TFs, suggestive of collaborative TF binding[2,29]. For example, the NKX2-5 motif was highly enriched at TBX5 regions, and the TBX5 motif was highly enriched at NKX2-5 regions (Fig. 1f). This finding was supported by central enrichment analysis (Supplementary Fig. 6b) and is consistent with the known biochemical and functional interactions between these factors[27]. Generally the seven cardiac TFs analyzed in this study enriched for each other's motifs more strongly than the motifs of other TFs (Supplementary Fig. 6a). Among these other TF motifs, those of MEIS and SMAD, TFs implicated in heart development, were the most strongly enriched (Supplementary Fig. 6a). The ETS motif was enriched at SRF regions, particularly in adult heart, in keeping

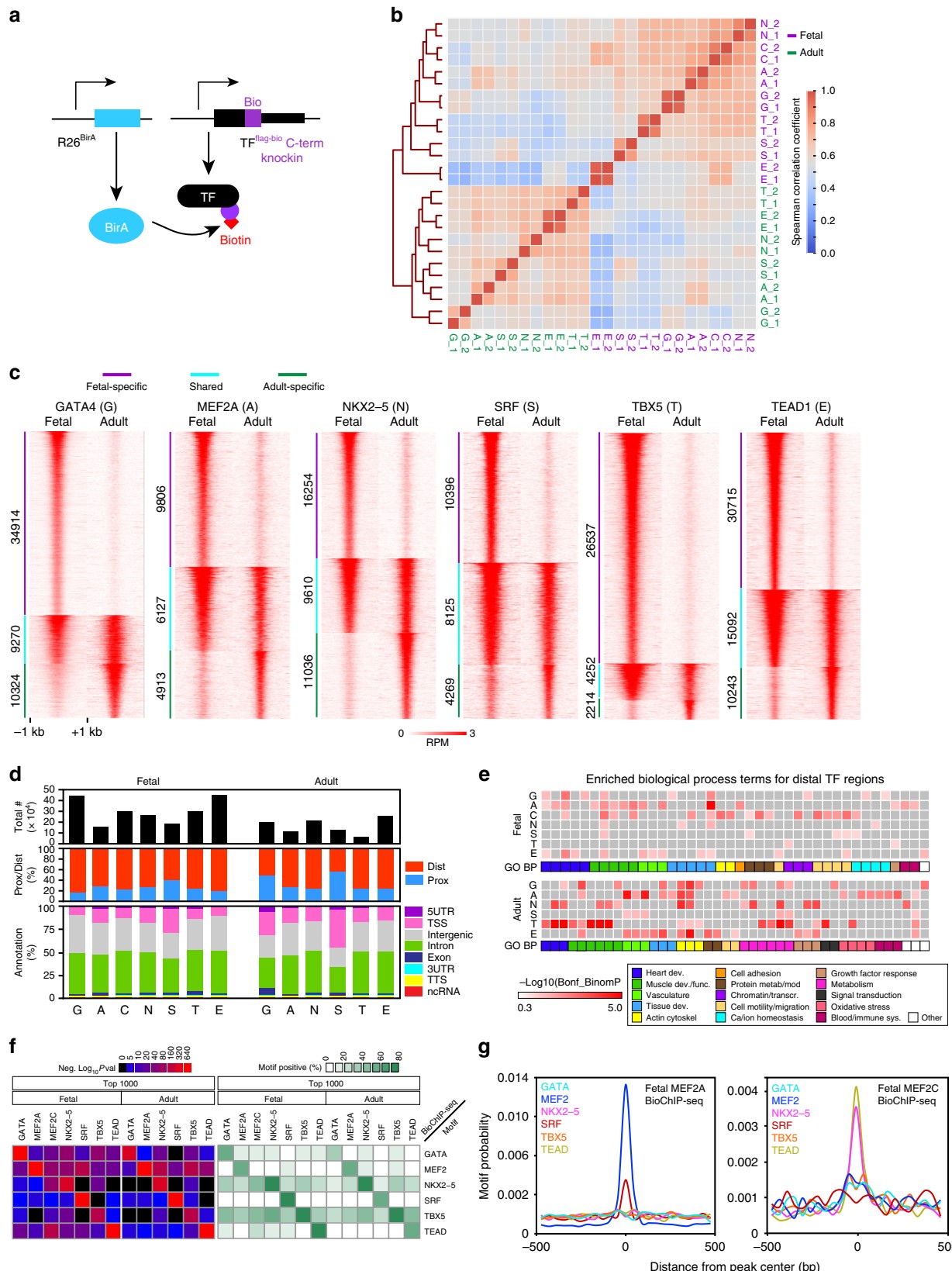

with cooperative DNA binding by SRF and the ETS family member ELK1[30]. Nuclear receptor motifs were not enriched amongst TF-bound regions in fetal heart, but showed enrichment in regions bound by several TFs in the adult heart (Supplementary Fig. 6a).

Together, bioChIP-seq robustly and reproducibly mapped the chromatin regions bound by seven different cardiac TFs at two different stages of heart development. These regions dynamically change during heart development. Motif analysis suggested significant collaborative interactions between TFs.

**Fig. 1** bioChIP-seq of major cardiac transcription factors in fetal and adult heart. **a** Strategy for bioChIP-seq. Murine knock-in alleles fuse a biotin acceptor peptide (BIO) to the C-terminus of target transcription factors (TFs). BirA expressed from the *Rosa26* locus modifies BIO with biotin, permitting high affinity pull-down under consistent conditions. **b** Correlation between aligned bioChIP-seq data obtained from heart ventricles. Fetal (F_; purple) and adult (A_; green) data were acquired in biological duplicates (_1 and _2). Heatmap shows Spearman correlation coefficients for signal within the union of peak regions across all replicates. G, GATA4; A, MEF2A; C, MEF2C; N, NKX2-5; S, SRF; T, TBX5; E, TEAD1. **c** Dynamic changes in TF binding between fetal and adult stages. Heatmaps of TF-bound regions, arranged into fetal-specific (purple), adult-specific (green), and shared groups (cyan). **d** Number and location of TF regions with respect to gene annotations. Middle row, regions proximal (within 2 kb) or distal (>2 kb) to the TSS. Bottom row, more detailed genome annotations, using definitions from Homer. "TSS" is defined as 1 kb upstream to 0.1 kb downstream of the TSS. ncRNA, non-coding RNA. See also Supplementary Data 1. **e** Enriched biological process gene ontology (GO) terms for genes neighboring distal TF-occupied regions, as defined by the default settings of GREAT[62]. The union of the five most significant terms for the top 1000 distal regions (ranked by BioChIP-seq signal) bound by each TF in fetal or adult stages. Color code indicates manual annotation of sets containing similar GO terms. Gray, non-significant *P*-value. See Supplementary Data 2 for the full table. **f** Motifs enriched among the top 1000 distal TF regions (summit ±100 bp). Regions were scanned for TF-binding motifs of the bioChIP'd TFs using Homer. Enrichment *P*-values (left) and percent positive regions (right) are shown. **g** Central enrichment plots for the top 1000 distal fetal MEF2A and MEF2C bioChIP-seq regions. MEF2A showed central enrichment for the MEF2 motif and weaker central enrichment for SRF. MEF2C showed central enrichment of TEAD, TBX5, and NKX2-5 that was greater than the MEF2 motif itself. Displayed motifs have *E*-value < $10^{-5}$. Source data for panels **d**–**g** are provided as a Source Data file

**Collaborative stage-specific TF chromatin occupancy**. To further investigate TF collaborative interactions, we analyzed the cobinding of TFs to the same genomic region (Fig. 2a and Supplementary Data 3). Multiple different TFs frequently bound to the same region. We further analyzed this co-occupancy by calculating the distance between adjacent peaks of all TF regions within each developmental stage. We focused on regions distal (>2 kb) to the TSS, to avoid potential effects of clustered TF binding near promoters. The inter-peak distance had a bimodal distribution (Fig. 2b), which was similar between intronic and intragenic locations (Supplementary Fig. 7a). The local maximum around $10^4$ bp approximates the expected inter-peak distance for randomly distributed peaks, whereas the peak at <300 bp (dotted red line, Fig. 2b) indicates substantial clustering of cardiac TFs. Based on this distribution, we defined a cobound region as one in which adjacent TF peak summits are no more than 300 bp apart. The majority (on average, 77%) of binding by each TF was to regions co-occupied by two or more different cardiac TFs (Supplementary Fig. 7b). Examples of TF co-occupancy are shown in Supplementary Fig. 7c–f. The median length of regions bound by five or more TFs was 702 and 590 bp in fetal and adult heart, respectively (Supplementary Fig. 7g). Regions co-occupied by multiple different TFs were more frequently found proximal to the TSS (±2 kb), particularly in the adult heart (Supplementary Fig. 7h). Indeed, the median distance from region centers to the TSS depended strongly on the number of co-bound TFs in adult but not fetal heart (Fig. 2c). This difference could not be accounted for by global differences in accessible chromatin distribution between developmental stages (Supplementary Fig. 8a).

Multiple TF co-occupancy of the same chromatin region ("multi-TF regions") suggests collaborative TF binding. Consistent with this[2,29], the TF bioChIP-seq signal increased with the number of different co-occupying TFs (Fig. 2d). Moreover, the accessibility of chromatin regions and the fraction of co-bound regions within accessible chromatin also increased with the number of different co-occupying TFs (Fig. 2e and Supplementary Fig. 8b), as predicted by the facilitated binding model in which TFs occupy a site by collaboratively displacing histones[29]. We also observed that regions with stage-specific TF co-occupancy were enriched for stage-specific chromatin accessibility (Supplementary Fig. 8c).

We further investigated co-occupancy relationships by examining the pairwise overlap between TF regions (Fig. 2f). In agreement with the motif analysis (Fig. 1f–g and Supplementary Fig. 6), fetal MEF2C and NKX2-5 bound regions extensively overlapped. Both MEF2C and NKX2-5 regions also frequently overlapped regions occupied by GATA4, TBX5, and TEAD1.

NKX2-5 co-occupancy with GATA4, TBX5, and TEAD1 was maintained in adult heart. Protein pull-down assays validated the interaction of TEAD1[fb] with endogenous MEF2C and NKX2-5 in fetal heart (Supplementary Fig. 9a). Similarly, NKX2-5[fb] interacted with MEF2C and TBX5 in fetal heart extracts (Supplementary Fig. 9b).

There were 128 and 64 possible TF co-occupancy patterns for the 7 fetal and 6 adult TFs. Some TF cobinding patterns were used more frequently than others, and the patterns changed between fetal and adult stages (Supplementary Fig. 10a–c). The most frequent cobinding patterns involved MEF2C, NKX2-5, and TEAD1 (Supplementary Fig. 10a–c). Individual TF cobinding patterns were associated with distinct gene ontology terms (Supplementary Fig. 10d; Supplementary Data 2).

Regions occupied by the top 20 most frequent TF co-occupancy patterns were assessed for DNA sequence motif enrichment (Supplementary Fig. 10e). The most significant motifs for each TF cobinding pattern corresponded to the bioChIP'd TFs, again with the exception of fetal MEF2C-occupied regions, which showed greatest enrichment for the NKX2-5 motif. Most regions co-occupied by ≥5 TFs in fetal or adult heart contained only 2–3 motifs of the analyzed TFs (Supplementary Fig. 10f), which were distributed along co-occupied regions rather than focally clustered (Supplementary Fig. 10g). These findings suggest that multi-TF region co-occupancy relies on protein–protein interactions, or TF binding at non-canonical motifs[31].

To further investigate motif orientation and spacing relationships, we analyzed regions co-occupied by each TF pair for the distance between TF motifs (Supplementary Fig. 10h). We did not observe a predominant motif distance for any TF pair. However, this analysis does not exclude that a small fraction of collaborative TF binding depends on fixed motif orientation and spacing. Therefore, we generated composite motif position weight matrices composed of all pairs of motifs in their four possible orientations and with 0–8 intervening bases (Supplementary Fig. 10i, top). We used these composite matrices to scan all regions co-occupied by each motif pair (Supplementary Fig. 10i–x). For the large majority of TF pairs, composite motif enrichment was not sensitive to motif arrangement, with some notable exceptions (Supplementary Fig. 10j). The most prominent exception was TBX5 and NKX2-5, which demonstrated preference for zero or four base pair motif spacing in an orientation specific manner (Supplementary Fig. 10i), consistent with the study of Luna-Zurita et al. of stem cell-derived cardiomyocytes[27]. The other TF pairs displaying the greatest variance of enrichment based on motif arrangement were: NKX2-5 and TEAD1

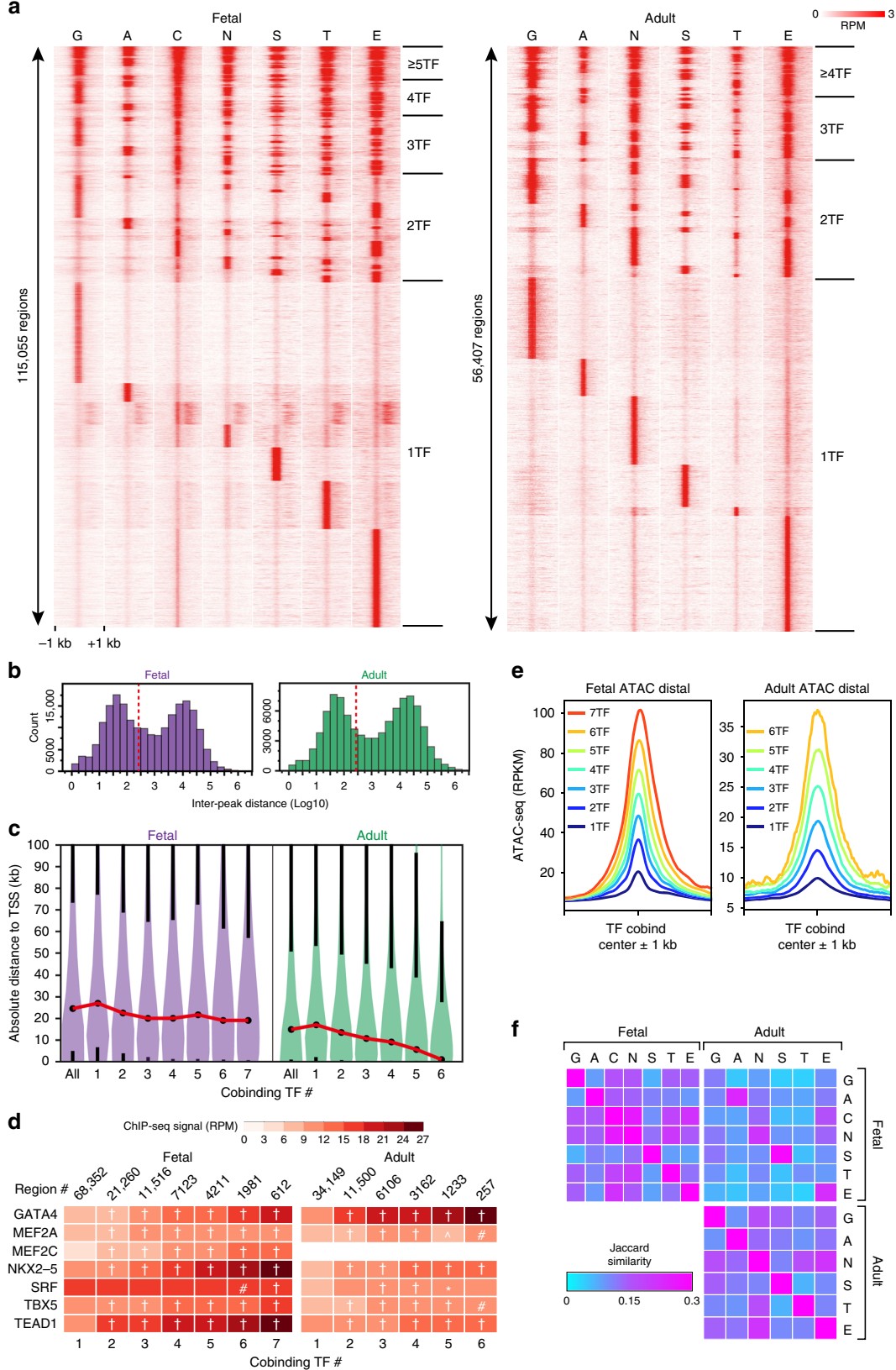

(Supplementary Fig. 10j, u), MEF2C and TBX5 (Supplementary Fig. 10j, r), and MEF2C and NKX2-5 (Supplementary Fig. 10j, p). No single motif orientation or spacing accounted for more than 5% of TF-bound regions, suggesting that the preferred arrangements contribute to a small fraction of overall TF binding.

These analyses showed that collaborative TF binding is highly prevalent genome-wide and identified specific sets of frequently interacting TFs, which differed between developmental stages. Most TF pairs co-occupy regions without fixed motif arrangement, consistent with a facilitated binding mechanism, such as

**Fig. 2** TF co-occupancy patterns in fetal and adult heart. **a** Heatmap of TF co-occupied regions, defined by merging individual TF peaks whose summits were within 300 bp of one another. The same region is shown in each row, with each column representing the signal of the indicated factor across the region. Regions are sorted by the pattern of co-occupying TFs. **b** Histogram of the distance between neighboring TF summits. TF regions within 2 kb of the TSS were excluded. Note the bimodal distribution. Dotted red line indicates 300 bp, the threshold used to define TF co-occupancy. **c** Violin plots showing distance of regions co-occupied by multiple TFs relative to gene TSSs. Black dots indicate median values, black whiskers start at the 1st and 3rd quartiles and extend 1.5 times the interquartile distance. The red lines connecting median values highlight the trend in the data. In adult heart, the number of co-occupying TFs was inversely related to distance between TF regions and TSS. **d** Relationship of number of cobinding TFs and mean TF bioChIP-seq signal. The bioChIP-seq signal, expressed as reads per million (RPM), within 200 bp regions centered on the MACS2-predicted summit of each TF within a TF-cobound region, is plotted as a heatmap. Steel non-parametric test vs. 1 TF: *, #, ^, † indicate $P < 0.05$, $P < 0.01$, $P < 0.001$, and $P < 0.0001$, respectively. **e** Average ATAC-seq signal in reads per kilobase per million reads (RPKM), centered at distal regions co-occupied by the indicated number of TFs. Chromatin accessibility, as measured by ATAC-seq, increased with increasing number of co-occupying TFs. **f** Pairwise co-occupancy (Jaccard Similarity) of TFs in fetal and adult heart (upper left and bottom right matrices), and across stages (upper right matrix). Note stage-specific differences in pairwise co-occupancy. Source data for panels **b**–**d**, **f** are provided as a Source Data file

collaborative histone displacement[29], whereas a small number of TF pairs have preferred motif arrangements suggestive of stabilizing ternary TF–TF–DNA interactions, as described recently for TBX5-NKX2-5[27].

**Relationship of TF-bound regions to H3K27ac-marked enhancers**. We characterized the location of TF-occupied regions with respect to H3K27ac, an epigenetic mark often used to identify transcriptional enhancers[32,33]. We utilized previously published H3K27ac data from fetal (E11.5) and adult mouse heart, liver, and forebrain[33]. Heart H3K27ac ChIP-seq showed strong signal at heart TF peak centers, and forebrain and liver H3K27ac was weaker (Fig. 3a). There was significant overlap between heart and non-heart H3K27ac regions, especially in the adult stage (Fig. 3b), such that adult heart H3K27ac sites did not enrich for cardiac functional terms (Supplementary Fig. 11a). To capture cardiac-enriched H3K27ac regions, we calculated a cardiac-selective H3K27ac score (CHS) for each region—its cardiac H3K27ac signal divided by the maximum of its liver and forebrain H3K27ac signal (Fig. 3c). The regions in the top two quintiles of CHS score (defined as cardiac H3K27ac regions, cHRs) mostly originated from heart samples (Fig. 3c) and were linked to relevant cardiac GO terms (Fig. 3d and Supplementary Data 2).

Next we analyzed the overlap of TF regions with H3K27ac regions. Only 16% of fetal and 62% of adult TF regions overlapped with H3K27ac regions from heart, liver, or forebrain (Fig. 3b). The extent of overlap between TF and H3K27ac regions increased with the number of co-occupying TFs, primarily due to greater overlap with regions with cHRs (Fig. 3e and Supplementary Fig. 11b). A substantial fraction (64% (fetal) and 33% (adult)) of regions co-occupied by ≥ 5 TFs did not overlap with cHRs (Supplementary Fig. 11c). Indeed, most cHRs in the adult heart were not bound by the studied TFs.

In contrast, the ATAC-seq signal at TF regions was strongly dependent on the number of co-occupying TFs (Fig. 2e), the H3K27ac signal strength showed weaker (adult) or no (fetal) relationship to the number of co-occupying TFs (Supplementary Fig. 11d). At TF regions where H3K27ac was below the statistical threshold to call H3K27ac peaks, weak H3K27ac signal remained detectable above random background. This sub-threshold H3K27ac signal was on par with heart H3K27ac signal at regions with "forebrain-specific" or "liver-specific" H3K27ac occupancy. We refer to these regions with subthreshold H3K27ac signal as "H3K27ac negative".

These data indicate that multi-TF binding increases the likelihood of a region bearing cardiac selective H3K27ac, but many multi-TF and H3K27ac regions do not overlap.

**Transcriptional enhancer function of TF regions**. Having identified regions co-occupied by multiple TFs with and without H3K27ac, we next sought to evaluate their biological function. First, we examined evolutionary conservation. Recent studies found that heart and liver enhancers, identified by H3K27ac or EP300 occupancy, exhibit lower conservation compared to analogous forebrain regions[33,34]. We observed the same weaker conservation of heart/liver H3K27ac regions (Fig. 4a). In contrast, heart TF region conservation was greater than heart H3K27ac regions (Fig. 4a), and exceeded that of forebrain H3K27ac regions in adult tissues. Regions occupied by greater numbers of TFs had higher conservation (Fig. 4b), with regions bound by ≥3 TFs having comparable or greater conservation than forebrain H3K27ac regions.

Quantitative analysis of mean conservation scores (Fig. 4c) supported three general observations. First, regions with greater numbers of co-occupying TFs showed greater conservation. Second, TF regions that did not overlap with heart H3K27ac were relatively well conserved. Third, overlap with heart H3K27ac slightly but significantly increased conservation of TF regions.

Second, we directly measured enhancer activity in cardiomyocytes in vivo using an adeno-associated virus (AAV)-based assay. We developed an AAV vector in which the hsp68 minimal promoter drives expression of an mCherry reporter, and the test enhancer is positioned within the 3′ UTR[35] (Fig. 4d). Four test enhancers were selected from each of the following three region classes from adult heart: (1) H3K27ac+ TF−, (2) H3K27ac− ≥5TF+, and (3) H3K27ac+ ≥ 5TF+. AAV containing individual enhancers was delivered to newborn mice, and enhancer activity was assessed at postnatal day 28 by measuring ventricular mCherry mRNA levels. To account for transduction efficiency, results were normalized to U6-driven Broccoli[36] internal control RNA, expressed from the AAV vector (Fig. 4d, e). ≥5TF regions with or without H3K27ac overall had stronger enhancer activity than H3K27ac+ TF− regions (Fig. 4e). Qualitative assessment of mCherry fluorescence corroborated these results (Fig. 4f and Supplementary Fig. 12a). A region adjacent to ryanodine receptor 2 (*Ryr2*) give divergent results in the RNA and fluorescence assays, likely because it was most active in atria, whereas RNA was measured from ventricles.

We expanded the enhancer assay to permit the parallel measurement of many enhancers[35,37–40]. We created an AAV library containing 2700 candidate regions, each 400 bp in length, cloned into the reporter gene's 3′ UTR[35] (Supplementary Fig. 12b). Test regions were selected to represent the three region classes above, plus negative controls, which consisted of regions occupied by P300 in embryonic stem cells[4] and H3K27ac forebrain enhancers (VISTA enhancer database[41]). The pooled

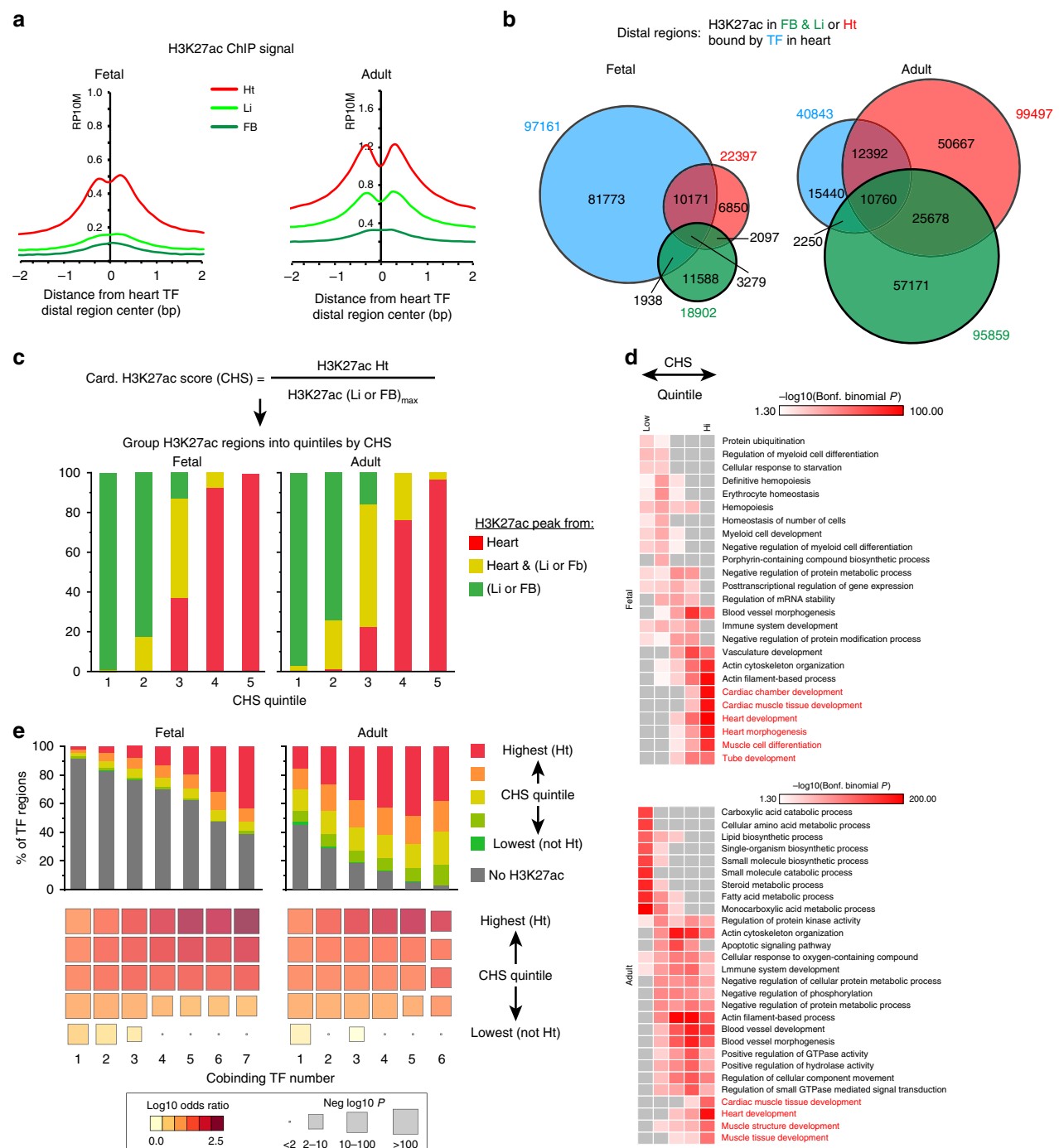

**Fig. 3** Relationship of TF and H3K27ac occupancy. **a** Average plot of H3K27ac ChIP-seq signal from heart, liver, or forebrain at heart distal TF summits. RP10M, reads per 10 million reads. **b** Overlap of distal H3K27ac regions in heart (Ht), forebrain (FB), and liver (Li) with distal heart TF regions. **c** Distal regions with cardiac-enriched H3K27ac occupancy. Distal regions were ranked by their cardiac H3K27ac score (CHS), which was the ratio of the H3K27ac signal in heart to the maximum signal in either forebrain or liver. The highest two quintiles were designated as cardiac H3K27ac regions (cHRs). **d** Biological process gene ontology terms for distal H3K27ac regions by CHS quintile. The top five terms for each quintile are shown. Red text highlights heart-related terms. Gray boxes indicate non-significant enrichment P-values. **e** Overlap of distal TF co-occupied and H3K27ac regions, stratified by CHS quintile and number of co-occupying TFs. In the lower bubble plot, color signifies odds ratio, and size denotes −log10(Fisher test P-Value). Source data for panels **c**–**e** are provided as a Source Data file

AAV library was delivered to newborn mice, and ventricular RNA was collected one week later. An amplicon containing the cloned enhancers was amplified by RT-PCR and analyzed by next-generation sequencing, so that the sequence of each region acted as its own barcode. Enhancer strength (RNA reads) was normalized to enhancer frequency in the overall library (AAV genomic DNA reads; Fig. 4g). Negative control regions had the lowest activity levels, and individually validated enhancer regions (Fig. 4d–f) generally agreed with the massively parallel measurements. This massively parallel reporter assay (MPRA) demonstrated that as a group H3K27ac⁺ TF⁻ regions were not sufficient to drive significant reporter activity, whereas ≥5TF regions were

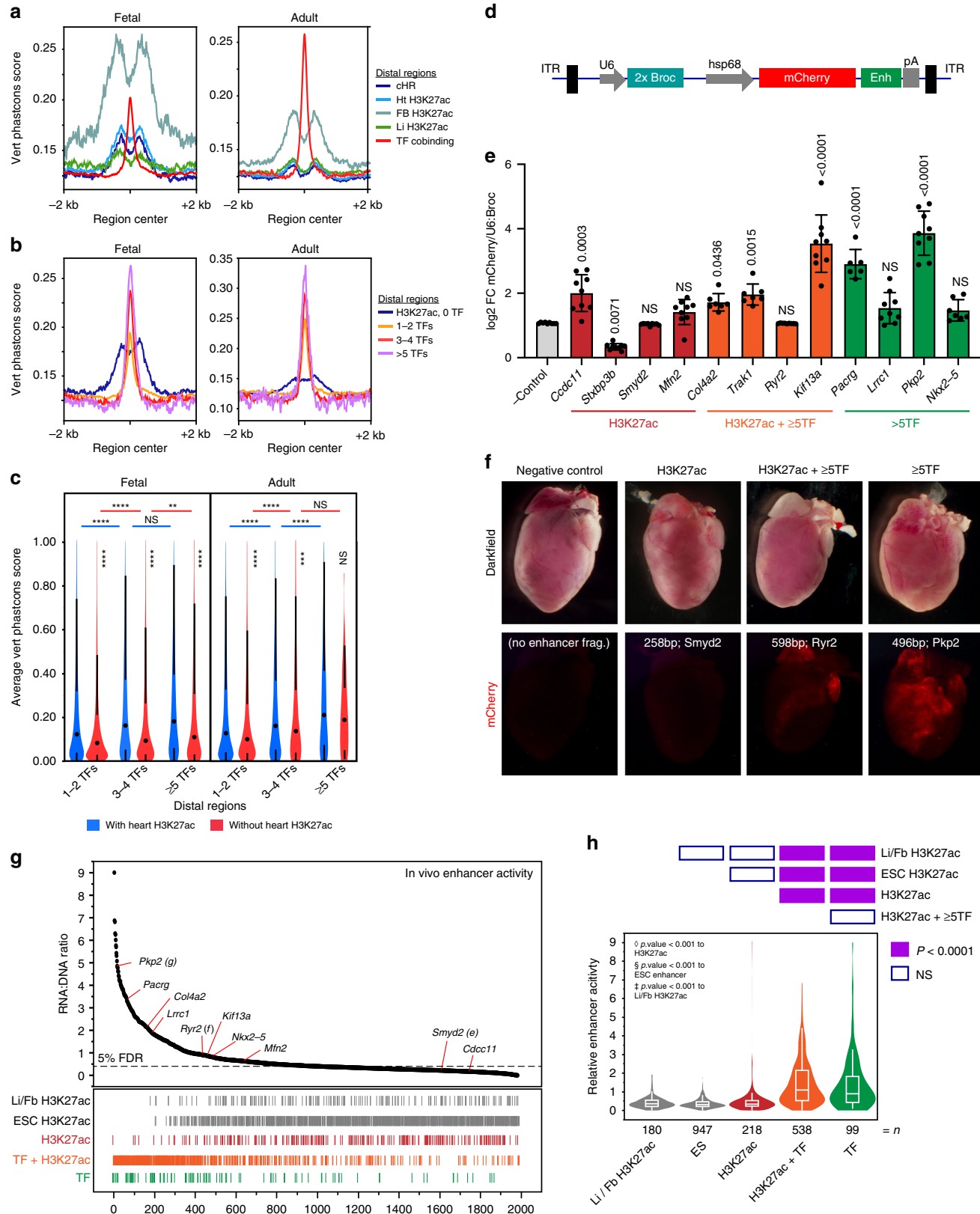

sufficient. Presence or absence of H3K27ac did not significantly affect enhancer activity (Fig. 4h).

Third, we searched the literature to identify previously reported cardiac enhancers that had been validated by transient transgenesis. We identified 52 enhancers (Supplementary

Fig. 12c and Supplementary Table 3). Regions containing greater numbers of co-bound TF were highly over-represented (≥35-fold for ≥5 TFs; Supplementary Fig. 12d), compared to random expectation. Regions occupied by H3K27ac were also highly enriched.

**Fig. 4** Transcriptional enhancer function of TF regions. **a** Average plot of 30-way vertebrate Phastcons scores at distal H3K27ac and TF regions. **b** Average plot of Phastcons scores at distal regions co-occupied by the indicated number of TFs, or by distal heart H3K27ac regions that do not overlap TF regions. **c** Average Phastcons scores in distal regions (center ± 150 bp) co-occupied by the indicated number of TFs with or without heart H3K27ac in fetal or adult heart. Steel-Dwass: ****$P < 0.0001$; ***$P < 0.001$; **$P < 0.01$; NS, not significant, mean values represented by dot (black). **d–f** Individual enhancer activity in murine cardiomyocytes was measured using an AAV9-based assay. Vector construct (**d**) contained minimal hsp68 promoter driving mCherry, with enhancer element (Enh) located in the 3′ UTR. U6:2xBroccoli was used to normalize transduction efficiency. Twelve enhancers belonging to three groups (H3K27ac$^+$ TF$^-$; H3K27ac$^+$ ≥5TF$^+$; H3K27ac$^-$ ≥5TF$^+$) were individually tested in vivo. Normalized enhancer activity was compared to empty vector (AAV9;Hsp68:mCherry-no enhancer; $n = 9$) using Dunnett's test. Error bars represent s.d. **e** Representative dark field and fluorescent wholemount images of adult (P28) mouse hearts showing enhancer activity for each group of cardiac enhancers. **g, h** AAV-based massively parallel reporter assay measurement of the activity of 1982 candidate enhancers and negative controls in cardiomyocytes in vivo. AAV library was delivered at P0 and measured in ventricles at P7. **g** Activity, expressed as enhancer reads in RNA normalized to enhancer abundance in AAV genomic DNA, was plotted versus region activity rank. Relative rank of individual regions of each class is represented by vertical lines below plot. 5% false discovery rate (FDR) threshold (dotted line) indicates the 95th percentile of ESC H3K27ac negative control region activity. **h** Relative activity of each class of regions tested is summarized displaying mean (line) and 95th confidence interval (box) with s.d. (whiskers). Heatmap above violin plot shows statistical significance of inter-group comparisons (Steel-Dwass). Source data for panels **c**, **e**, **g** are provided as a Source Data file

Together these data show that TF-bound regions are active enhancers, with or without H3K27ac cobinding. Multi-TF regions constitute a class of relatively well conserved cardiac regulatory elements that differ from previously reported, weakly conserved heart enhancers[33,34].

### Integration of multiple features to identify cardiac enhancers.
Chromatin features (open chromatin or occupancy by H3K27ac, p300, or TFs) have been used to identify enhancers[15,33,42]. To understand how these different chromatin features can be integrated to optimally predict cardiac enhancers, we used machine learning in combination with the Vista Enhancer database[41], in which 1044 murine sequences were tested for enhancer activity in E11.5 embryos by transient transgenesis, and 158 and 167 exhibited cardiac or forebrain activity, respectively. Among individual cardiac chromatin features examined (individual TF occupancy; H3K27ac; ATAC-seq; number of co-occupying TFs), ATAC-seq had the highest recall (sensitivity) but the lowest precision (Fig. 5a) for heart enhancer activity. Individual TFs had lower recall and higher precision, and H3K27ac was intermediate for recall and precision. Single TF regions had low recall and low precision, whereas multi-TF regions had intermediate recall and high precision. Chromatin feature predictions were tissue specific, as heart chromatin features did not perform well for prediction of forebrain enhancer activity (Supplementary Fig. 13a). Varying the signal threshold used for feature detection yielded receiver-operator curves that describe each feature's classification performance across a range of false positive rates (Fig. 5b and Supplementary Fig. 13b). The area under the resulting curve (AUC), a measure of classification accuracy, is summarized for several chromatin feature predictors in Fig. 5b. Among these individual features, the number of co-bound TFs had the highest classification accuracy. Again, the cardiac chromatin features had much higher predictive accuracy for heart enhancer activity compared to forebrain (Supplementary Fig. 13c).

We used machine learning (ML) to develop an ensemble decision tree-based classifier of active heart enhancers that integrates information from multiple chromatin features. Classifier performance, evaluated by five-fold cross-validation, showed that its AUC score was $0.8815 \pm 0.0036$ (mean ± sd; Fig. 5c), which is higher than any individual feature. Examining the relative importance of each feature for ML classifier performance indicated that the number of co-occupying TFs and H3K27ac were the top two features (Fig. 5d). Repeating the analysis without either H3K27ac, the number of co-occupying TFs, or both showed that omission of the number of co-occupying TFs impaired classification accuracy, but omission of H3K27ac did

not (Fig. 5e), suggesting that for purposes of classification H3K27ac is redundant with other model features.

These data indicate that TF occupancy and the number of co-occupying TFs, in addition to H3K27ac, are important features for classification of active enhancers.

### TF regions regulate developmental gene expression.
We next investigated the function of TF-bound regions in regulating gene expression. First, we analyzed gene expression in normal fetal and adult cardiomyocytes (Supplementary Table 3 and Supplementary Data 4). Genes were annotated by the regions within 100 kb of the TSS with the highest number of co-occupying TFs, and by the presence of overlapping heart H3K27ac at these regions (see the Methods section). Genes associated with regions with greater cobinding TF number had higher expression in both fetal and adult (Fig. 6a). For regions co-occupied by TFs and H3K27ac, the greatest effect on gene expression occurred between zero and 1 TF, with the addition of more cobound TFs having diminishing effect. For regions occupied by TFs but not H3K27ac, there was a more graded effect of increasing cobinding TF number. Genes associated with the same number of cobound TFs had higher expression with H327ac co-occupancy than without, except at the highest numbers of cobound TFs (Fig. 6a). These data suggest that both the number of cobinding TFs and H3K27ac co-occupancy impact overall gene expression level.

We analyzed the association of TF or H3K27ac regions with genes differentially expressed between fetal and adult heart (absolute $\log_2$fold-change > 2 and $p < 0.001$; Fig. 6b). TF or H3K27ac regions were enriched adjacent to genes with adult-biased or fetal-biased expression (Fig. 6c). This enrichment was greater for regions with H3K27ac compared to those without. In adult but not fetal heart there was a positive correlation between number of co-bound TFs and the degree of enrichment.

Altogether, these results implicate regions occupied by TFs and H3K27ac in regulating gene expression level and temporal specificity.

### TEAD1 is a key regulator of cardiac gene expression.
TEAD1 has become known as a major nuclear target of Hippo-YAP signaling that regulates cell proliferation. However, earlier literature[13,15,43] implicated TEAD1 as an integral participant in cardiac gene regulation, and this was further supported in this study. We analyzed differentially expressed genes (DEGs) resulting from stage-specific *Tead1* inactivation[44] in heart with respect to regions co-occupied by TEAD1, other TFs, and H3K27ac (see the Methods section; Fig. 7a and Supplementary Data 4). Regions co-occupied by TEAD1 and other bioChIP'd

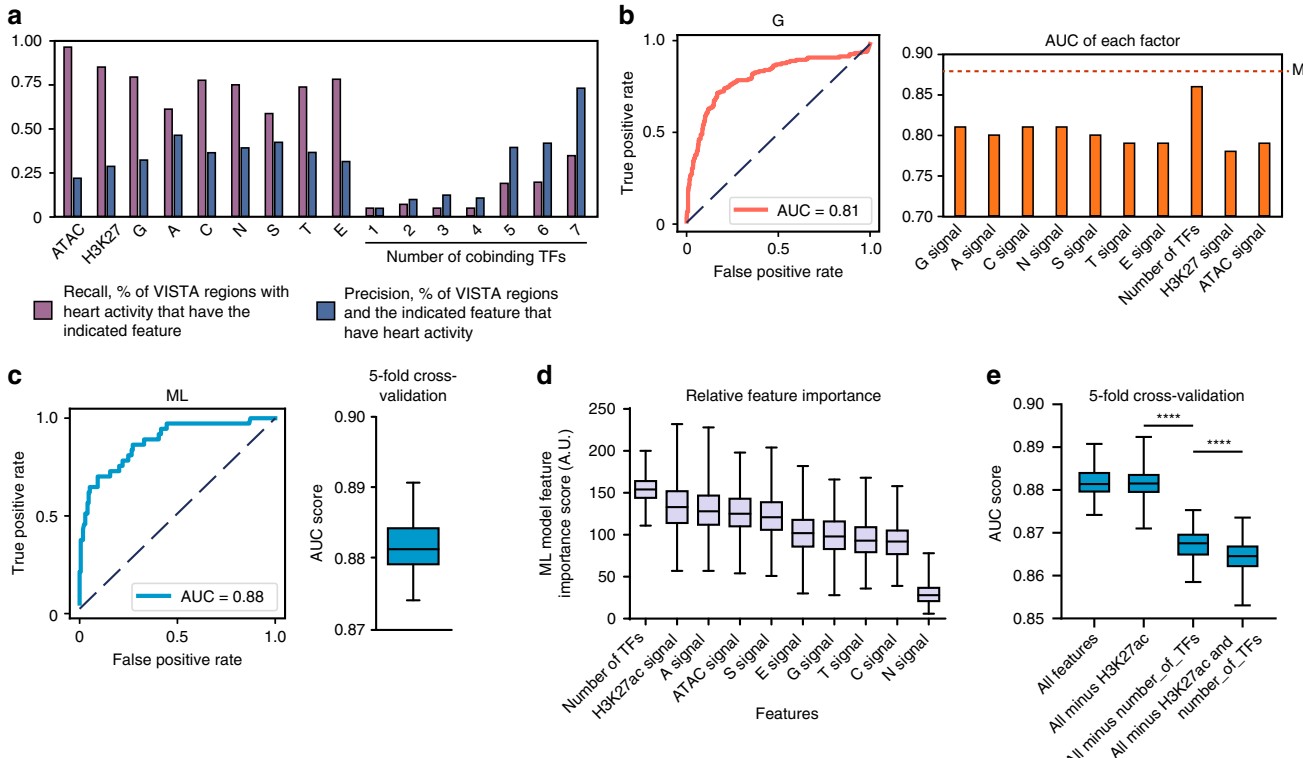

**Fig. 5** Multi-TF co-occupancy contributes to prediction of active heart enhancers. The VISTA enhancer database[41] was used as the gold standard to develop classifiers of transcriptional enhancers based on TF co-occupancy, H3K27ac occupancy, and chromatin accessibility. **a** Recall and precision of VISTA heart enhancer activity based on heart chromatin features. **b** VISTA heart enhancer receiver-operating characteristic curve and area-under-the-curve (AUC) scores for each factor. An example ROC curve for enhancer prediction by fetal GATA4 bioChIP-signal is shown in the left panel. Similar analyses of other chromatin features is shown in Supplementary Fig. 13. The right plot summarizes the predictive accuracy, assessed by the AUC, for the indicated chromatin features. **c** Machine learning (ML) classifier performance for classification of VISTA heart enhancers. An ensemble decision tree-based classifier was trained using 80% of the data and tested on the remainder. Classifier performance receiver-operating characteristic curve (left) yielded AUC = 0.88. Performance in 100 permutations of five-fold cross-validation is shown by the box plot to the right. The average ML classifier AUC is indicated by the red dashed line in panel **b**. **d** Relative feature importance in the ML classifier. The box plots show the relative importance of individual chromatin features for classifier performance in 100 permutations of five-fold cross validation. **e** Effect of omission of top ranked classifier features on classifier performance. Omission of H3K27ac signal did not affect classifier performance, but omission of "number of TFs" impaired classifier performance. Box plots in figure; 1st through 3rd quartile represented by box and whiskers denoting min and max values, Source data for panels **a**-**e** are provided as a Source Data file

TFs were enriched adjacent to TEAD1-downregulated DEGs; enrichment was weaker and less significant adjacent to upregulated DEGs. H3K27ac co-occupancy increased enrichment of TF regions adjacent to DEGs. There was no clear relationship between the number of TFs that co-bound with TEAD1 and enrichment adjacent to DEGs.

Top functional terms for downregulated, TEAD1-bound DEGs in fetal heart related to myofibrils, muscle contraction, and sarcomeres, whereas in adult heart they related to mitochondrial and metabolic processes (Fig. 7b). Cell proliferation and tissue growth were notably absent among the top ranked terms.

Since our analysis revealed high co-occupancy between NKX2-5 and TEAD1 (Fig. 1f) and NKX2-5–TEAD1 biochemical interaction (Supplementary Fig. 9), we analyzed NKX2-5 region enrichment adjacent to TEAD1 DEGs (Fig. 7c, d). NKX2-5 regions, especially those co-marked by H3K27ac, were over-represented adjacent to TEAD1 down-regulated DEGs, but not adjacent to up-regulated DEGs, implicating NKX2-5–TEAD1 in target gene activation. Conversely, we also tested the hypothesis that TEAD1 contributes to regulation of genes downstream of NKX2-5 by analyzing several published *Nkx2-5* homozygous or heterozygous loss of function datasets (Supplementary Data 4).

For many but not all datasets, NKX2-5 and TEAD1 regions, especially those with H3K27ac co-occupancy, were enriched adjacent to genes downregulated in *Nkx2-5* loss-of-function models (Fig. 7c, d). The number of co-occupying TFs was not consistently related to region enrichment adjacent to DEGs.

Our motif analysis identified preferred arrangements of NKX2-5 and TEAD1 motifs in fetal co-bound regions (Fig. 7e and Supplementary Fig. 10j). We investigated whether these preferred arrangements were functionally significant by asking whether NKX2-5 and TEAD1 co-occupied regions containing this composite motif were enriched adjacent to *Tead1* or *Nkx2-5* DEGs in fetal heart, relative to co-occupied regions containing the non-preferred motifs arrangements (Fig. 7f). The preferred motifs were significantly over-represented adjacent to a subset of these DEGs, suggesting that these preferred motifs have increased regulatory activity.

Together, these analyses indicate that TEAD1 is an integral component of the cardiac transcriptional regulatory network and that it regulates core cardiomyocyte-specific functions including contraction and metabolism. TEAD1 and NKX2-5 physically interact, and occupy regions with a preferred motif orientation and spacing that is functionally significant.

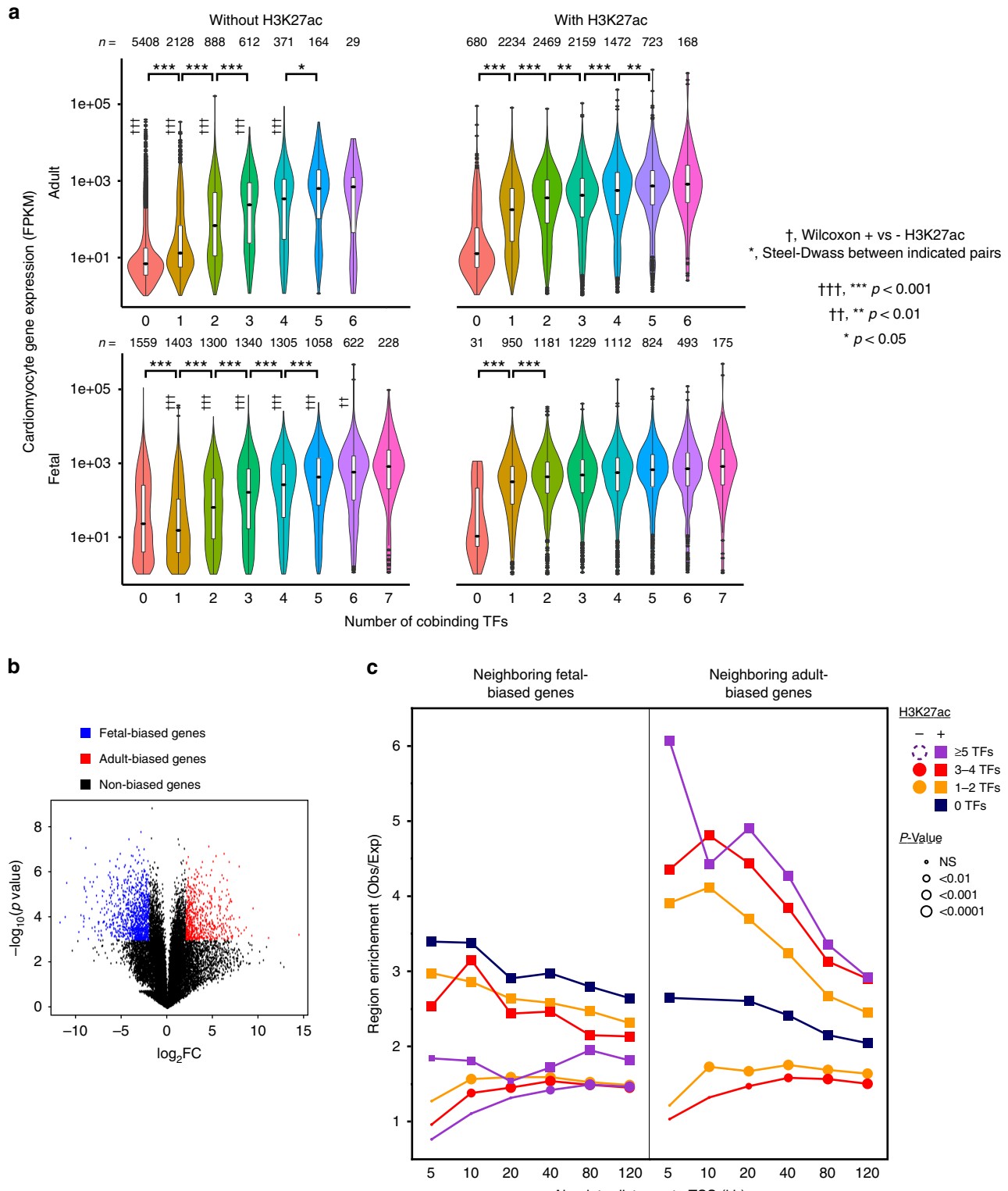

† , Wilcoxon + vs - H3K27ac
*, Steel-Dwass between indicated pairs

†††, *** p < 0.001

††, ** p < 0.01

* p < 0.05

## Discussion

Using bioChIP-seq, we established a reference map of TF occupancy for seven cardiac TFs at two stages of heart development. Binding of all cardiac TFs studied was highly dynamic between fetal and adult heart. This extends our prior observations on GATA4[3] and suggests that it is a general principle in the development of the heart and other organs. Cardiac TFs exhibited extensive collaborative binding, and detailed motif analysis suggests that collaborative binding is largely driven by indirect cooperativity. There were four TF pairs that did demonstrate preferred motif arrangements suggestive of cooperative binding through defined ternary complexes. Even in these cases, the preferred motif arrangement comprised a small fraction of all binding sites. One TF pair with preferred motif arrangement was NKX2-5 and TBX5, which matched the recent study of Luna-Zurita et al. and is consistent a defined protein–protein

**Fig. 6** Stage-specific TF chromatin occupancy regulates stage-specific gene expression. **a** Cardiomyocyte gene expression association with TF regions and heart H3K27ac. Genes were assigned to the region within 100 kb with the highest number of cobinding TFs. This region was classified as with or without H3K27ac overlap. †, Wilcoxon test for expression with heart H3K27ac compared to without for same TF number and stage. *, Steel-Dwass for expression difference between indicated groups. *n* indicates group size and vertical line denotes s.e.m. with outliers plotted. **b**, **c** Fetal or adult-based cardiac gene expression association with fetal-TF or adult-TF regions. Volcano plot (**b**) of fetal vs. adult cardiomyocyte gene expression shows fetal-biased genes (blue) and adult-biased genes (red), defined as $-\log_{10}(P$ value$) > 3$ and abs$(\log_2$FC$) > 2$. Right plot (**c**) shows enrichment (observed/expected) of TF or H3K27ac regions within the indicated distance of the TSS of fetal-biased or adult-biased genes. The expected number was determined by randomly permuting the TF or H3K27ac regions 10,000 times across the mappable genome (see the section Methods). The size of symbols indicates the permutation *P*-value. Squares and circles indicate regions with or without H3K27ac co-occupancy, respectively. TF co-occupied region sets containing <100 members were excluded (dashed symbol in figure legend). Source data for panels **a**, **c** are provided as a Source Data file

interaction interface that favors binding to specific motif arrangements[27]. A second involved NKX2-5 and TEAD1. NKX2-5 and TEAD1 co-occupied regions containing the preferred motif arrangements were overrepresented adjacent to DEGs, suggesting that the preferred motif arrangements promote formation of a stereotyped complex with distinct regulatory properties.

Transcriptional enhancers are often marked by H3K27ac and p300 occupancy[33,42]. Consistent with our prior in vitro studies of the HL1 cardiomyocyte-like cell line[15], we found that in vivo regions occupied by multiple TFs incompletely overlap with H3K27ac. Multi-TF regions were highly conserved and highly active in vivo in reporter assays, and H3K27ac co-occupancy had small to no overall effect on these functional measures. In contrast, cardiac H3K27ac regions without TF co-occupancy had weak conservation, as noted previously[33,34], and overall no detectable enhancer activity compared to negative controls. However, at the level of gene expression we observed that H3K27ac regions were enriched adjacent to DEGs, whereas there was no consistent pattern related to the number of co-occupying TFs. This difference suggests that the factors that determine whether an individual region is sufficient for enhancer activity are distinct from those that determine overall gene expression, which reflects the complex integration of inputs from multiple cis-regulatory regions.

Our analysis revealed that the chromatin occupancy patterns of MEF2A and MEF2C are very different, despite DNA-binding domains that are 99% identical. The distinct binding patterns of these factors is consistent with their non-redundant and distinct function in the heart[9,10]. We showed that MEF2C occupies a substantial fraction of its sites through NKX2-5 and TEAD1, consistent with the previously reported physical and genetic interactions of MEF2C and NKX2-5 in regulating formation of the ventricle[45]. Our study extends this work by illuminating the extensive interdependence of these factors for MEF2C chromatin occupancy in the fetal heart.

Although TEAD1 has not commonly been included among central cardiac TFs, it is required for heart development[13], and its motif is found at muscle gene promoters[43]. Our analysis indicates that TEAD1 has a specialized role in cardiac muscle to collaborate with other cardiac TFs, especially NKX2-5, to regulate cardiac gene expression, in addition to its role in multiple cell types as a target of Hippo/YAP growth signaling[12]. This dual nature of TEAD1 is similar to that of SRF, a broadly expressed TF that likewise has specialized functions in cardiomyocytes.

Our reference map of cardiac TF occupancy will be an important starting point for elucidating the transcriptional network that governs heart development, homeostasis, and disease responses. The current work was limited to ventricular tissue at two time points. Addition of similar high-quality chromatin occupancy data in other cardiac tissues (e.g. atria), at other developmental stages, in purified cell types, and in disease models will illuminate how transcriptional programs adapt to different cellular contexts and stimuli. The resulting resource will provide the foundation to dissect gene regulatory mechanisms in heart development and disease.

## Methods

**Mice.** Mouse husbandry and procedures were performed under the approval and observation of the Boston Children's Hospital International Animal Care and Use Committee. *Tbx5^fb^* were generated Frank Conlon. *R26^BirA 20^* mice were obtained from Jackson labs (Jackson #010920). *Gata4^fb^* (Jackson #018121)[15], *Srf ^fb^* (MMRRC #37511-JAX)[16], *Tead1^fb^* (MMRRC #037514-JAX)[18], and *Tbx5^fb/fb19^* mice, in which epitope tag consisting of FLAG and BIO sequences is knocked in at the stop codon of the endogenous gene, were described previously. The *Nkx2-5^fb^* allele (Jackson #025978) was generated by homologous recombination in murine embryonic stem cells as described previously[17]. *Mef2a^fb^* and *Mef2c^fb^* (Jackson #025983) alleles were generated by Cas9-stimulated homologous recombination in murine embryonic stem cells. Flp-flanked selection cassettes were removed by mating to Flp-expressing mice (Jackson #016226)[46] and then breeding out the Flp allele. Details for these three newly reported mouse lines are provided in Supplementary Fig. 1. All subsequent mice were maintained in a mixed genetic background. Primers are listed in Supplementary Table 4. *Tead1^flox^,SM22a-Cre*, and *CAGGCre-ER* mice were reported previously[47–49].

**Echocardiography.** Echos were performed using a VisualSonics Vevo 2100 machine with Vevostrain software. Animals were conscious during procedure and the echocardiographer was blinded to the genotype of each animal.

**AAV reporter assays.** Genomic regions were selected based on region co-occupancy by ≥5TFs with strong ChIP-seq signal for each TF, and/or strong H3K27ac ChIP-seq signal combined with high expression of adjacent gene in adult heart. Regions were PCR amplified using primers listed in Supplementary Table 4 and cloned into the ITR-containing AAV plasmid. Adeno-associated virus serotype 9 (AAV9) was generated by transfecting HEK293T cells using polyethylenimine (PEI), the ITR-containing enhancer reporter constructs, and appropriate helper plasmids. Virus was harvested 72 h post transfection, purified using OptiPrep density gradient purification (Sigma), and concentrated with a 100 kDa Amicon Ultra Centrifugal Filter (Millipore). Purified virus (1.25E^12^ viral genomes/pup) was diluted and 50 µl subcutaneously injected into newborn pups (P0). Hearts were harvested at P28. RNA was extracted using TRIzol (Life Technologies) and purified with Zymo RNA Clean and Concentrator kit. Reporter activity was determined by performing qRT-PCR analysis for mCherry and dimeric Broccoli (Addgene #66845) transcripts.

**Massively parallel reporter assay (MPRA).** Enhancers were synthesized by Agilent as an oligonucleotide pool. Each enhancer consisted of two 230 bp ssDNA oligos with 20 bp 3′ overlap. Each oligonucleotide's 5′ end had a 20 bp primer-binding site. Oligonucleotides within the pool were annealed and 3′ ends were extended by PCR to create a library of 400 bp enhancers flanked by 20 bp priming sites. NotI/AscI-restriction sites were added to enhancers in a second round of PCR. The enhancer library was then digested, size selected, and ligated into the multiple cloning site of a self-complementary AAV plasmid containing a minimal MLC2v promoter-mCherry-NotI/AscI-polyA sequences. The ligation product was electroporated into Agilent SURE Electrocompetent cells following manufacturer recommendations, spread onto agar plates, and cultured overnight. Approximately 900,000 colonies were harvested and pooled for plasmid maxi-prep. The enhancer library was packaged into AAV as described above. P0 wildtype CFW mouse pups (*n* = 28) were injected subcutaneously with 50 µl saline containing containing 2E11 vg. Hearts were harvested at P7 and RNA was isolated from homogenized ventricular apexes via TRIzol phase extraction and reverse transcribed using a primer recognizing the start of the polyA sequence. NGS adapters and unique indexes were added to each sample in subsequent rounds of PCR amplification. Untransduced AAV DNA from the library pool was also prepared in the same fashion for sequencing in triplicate. Indexed samples were pooled and paired-end (2 × 150 bp) sequenced on a NextSeq500. After removal of adapters, reads were aligned to the mouse genome, keeping only mates that produced concordant alignments between

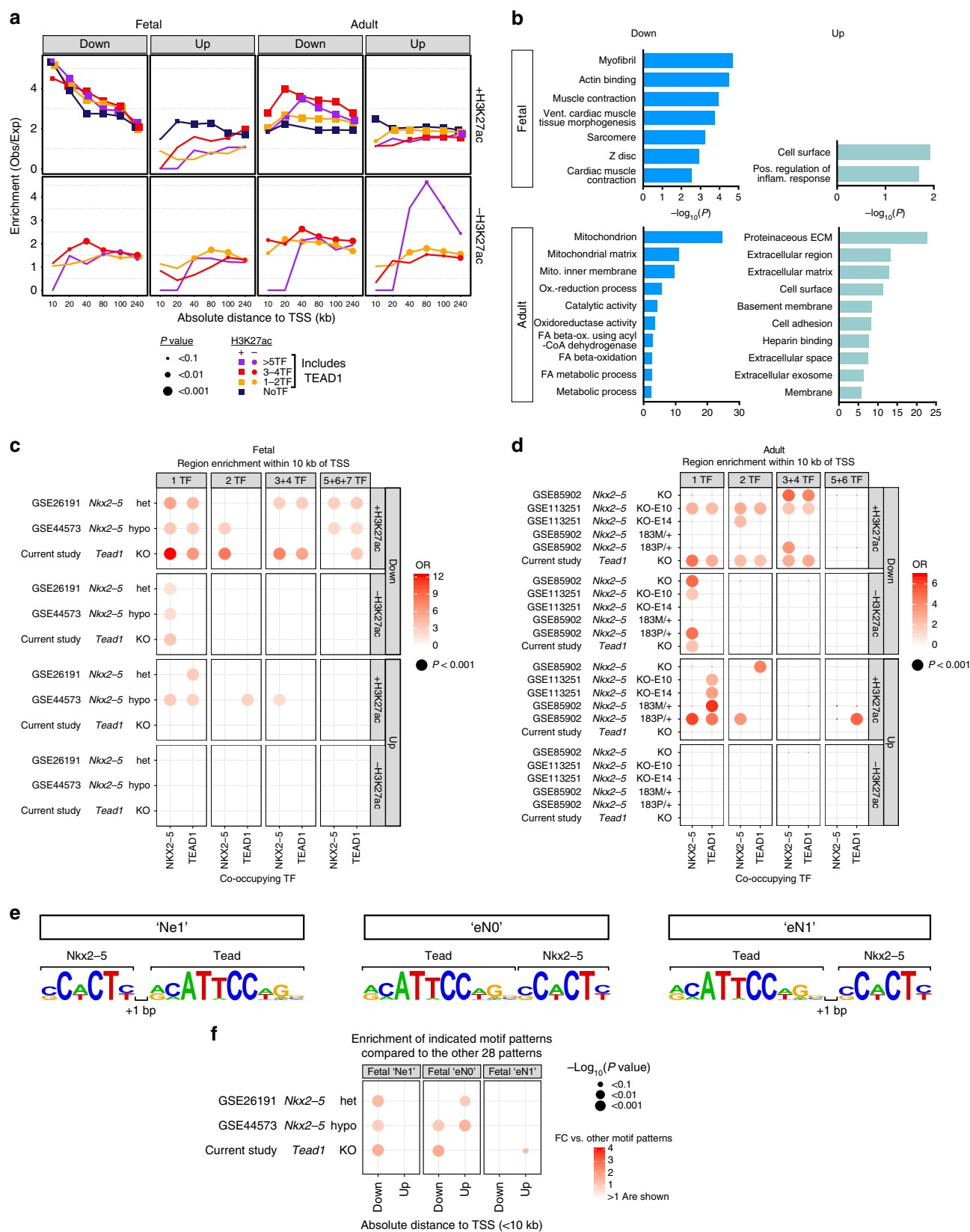

395 and 405 bp. On average, each sample contained ~5M alignments passing these criteria. The number of reads for each enhancer in each sample was determined using BedTools[50]. The average number of reads for each enhancer within the untransduced AAV DNA was then calculated, and enhancers present at low frequencies (<5 RPM) were excluded. The majority (>90%) of enhancers were successfully created and were detected above the 5 RPM threshold. Read numbers for

RNA samples were acquired using the same method. RNA:DNA ratio for each region was calculated and used as a readout of enhancer activity.

**Tissue harvest and chromatin precipitation.** Tissues were harvested in ice cold 1% formaldehyde/PBS from embryonic day 12.5 (E12.5) and adult (P42) transgenic

**Fig. 7** TEAD1 is an integral regulator of cardiac gene expression. **a** Enrichment of TF or H3K27ac regions adjacent to genes up-regulated and down-regulated in fetal or adult *Tead1* knockout hearts. TF regions contained TEAD1. Enrichment (observed/expected) of TF or H3K27ac regions within the indicated distance of differentially expressed genes' TSS was calculated by permutation testing. Expected values were determined by randomly distributing regions across the mappable genome in 3000 permutations. Size of symbols indicate permutation test *P*-value. TEAD1-occupied regions marked by H3K27ac were enriched adjacent to upregulated DEGs. **b** GO term analysis of genes down-regulated or up-regulated in TEAD1 knockout and adjacent to TEAD1-bound regions. Down-regulated, TEAD1-bound genes were enriched for biological process terms related to cardiomyocyte-specific functions. **c**, **d** Bubble plots indicating enrichment of regions occupied by the indicated TF (NKX2-5 or TEAD1) within 10 kb of the TSS of up-regulated or down-regulated genes in the indicated datasets involving either *Tead1* knockout or *Nkx2-5* mutation. NKX2-5 or TEAD1-bound regions marked by H3K27ac were enriched adjacent to genes downregulated by *Nkx2-5* or *Tead1* mutation. **e** "Preferred" arrangements of NKX2-5 and TEAD1 motifs in regions co-bound by these TFs (see Supplementary Fig. 10j). **f** Enrichment of regions co-bound by NKX2-5 and TEAD1 and containing the indicated motif arrangement adjacent to differentially expressed genes in fetal heart. Bubble size indicates permutation *P*-value (3000 permutations). Color indicates fold-change of enrichment compared to co-bound regions containing the 28 non-preferred motif arrangements. Source data for panels **a**, **c**, **d**, **f** are provided as a Source Data file

animals for all chromatin immunoprecipitations. All bioChIP-Seq data presented was generated from heterozygous TF bio-tagged animals (mixed strain backgrounds). Fetal transgenic heart ventricles were dissected from atria and cardiac cushions. Adult ventricle apexes (distal 1/3) were dissected away from remaining heart tissues. Homogenized tissues were crosslinked in fresh 1% formaldehyde/PBS for 30 min, rotating at room temperature and subsequently quenched with 500 mM glycine for 5 min. Crosslinked samples were centrifuged and washed multiple times with cold PBS. Crude nuclei preparation was performed using a hypotonic buffer (+protease inhibitors, 1 mM DTT), before snap-freezing in liquid nitrogen and stored at −80 °C. Pooled embryonic litters (~100 hearts) and between 2 and 3 female and male adult mouse littermates (4–6 total) were used for each bioChIP-Seq replicate experiment. Crude nuclei were thawed on ice and lysed again using a mild hypotonic buffer (20 mM Hepes pH 7.5, 10 mM KCl, 1 mM EDTA, 0.1 mM active Na$_3$VO$_4$, 0.5% NP-40, 10% glycerol, 1 mM DTT, Roche cOmplete protease inhibitors) and douncing with a glass 'B' pestle 20–30 times. Crosslinked nuclei were sonicated in 2.5 ml ChIP dilution buffer (0.25% SDS) using a QSonica 700 with microtip probe (cat. 4417). Sonicated chromatin was centrifuged, and the supernatant used for chromatin immunoprecipitation (ChIP) with Invitrogen M280 Streptavidin DynaBeads (Life Technologies). Chromatin immunoprecipitations were performed for 4 h, rotating, at 4 °C and washed extensively with buffers containing protease inhibitors, at room temperature, 5′ minutes each. Chromatin/beads were resuspended in SDS elution buffer (1% SDS, 10 mM EDTA, 50 mM Tris–HCl pH 8) and incubated overnight in a 65 °C water bath to reverse crosslink and elute from Dynabeads. DNA was RNAse A and Proteinase K treated the following day and purified using Qiagen MinElute (cat. 28004) for subsequent NGS library synthesis.

**Protein analyses.** Tissue harvested for protein expression was immediately placed in cold RIPA buffer (10 mM Tris–Cl pH 8, 1 mM EDTA, 1% Triton X-100, 0.1% sodium deoxycholate, 0.1% SDS, 140 mM NaCl, 1 mM DTT) + protease inhibitors. Samples were homogenized and debris pelleted. Protein concentration was determined using BCA Protein assay kit (Thermo Fisher). [0.5 µg/µL] lysates were used to test expression (Supplementary Fig. 1) and input controls for CoIPs (Supplementary Fig. 9). CoIP assays were performed with 150 µg embryonic heart lysate (Nkx2-5$^{fb/fb}$;R26$^{BirA/BirA}$) for 4 h at 4 °C. M280 Dynabeads were gently and thoroughly washed with cold 1X PBS (+Protease inhibitors). Beads were resuspended in 25 µl RIPA for subsequent protein assays. Proteins were separated and quantified by capillary electrophoresis (ProteinSimple Wes instrument). Antibodies and dilutions are listed in Supplementary Table 5.

**RNA isolation, RNA-Seq library preparation, and analysis.** Fetal Tead1 KO/WT heart samples consisted of E13.5 hearts and dorsal aorta from *SM22a:Cre;Tead1 flox/flox; Rosa26$^{mTmG}$/+* and *SM22a:Cre;Tead1 flox/+;Rosa26$^{mTmG}$/+* respectively[44]. Tissues were dissociated and GFP+ cells were isolated by flow cytometry for RNA isolation. Adult *Tead1* KO/WT heart samples (~10 hearts) were generated from *CAGGCre-ER;Tead1 flox/flox* and *CAGGCre-ER;Tead1$^{+/+}$* hearts. Samples were collected for RNA extraction, libraries prepared and sequenced and reads aligned as previously described[44]. Differentially expressed genes were identified using raw gene counts with the Bioconductor package EdgeR[51].

For normal fetal and adult cardiomyocyte expression data, isolated, purified cardiomyocytes were used with three biological replicates per group. Adult CMs were dissociated and purified from 6-week-old hearts by Langendorf collagenase perfusion followed by differential sedimentation as described[16]. Fetal hearts were dissociated using the Neonatal mouse cardiomyocyte isolation kit (Cellutron Life Technologies) and purified using the mouse neonatal cardiomyocyte isolation kit (Miltenyi). CM preparations were over 90% pure by fluorescence microscopy. RNA was purified using the Purelink RNA mini kit (Life Technologies). Ribosomal RNA was depleted using Ribo-Zero rRNA removal kits (Epicentre). RNA-seq libraries were prepared using ScriptSeq v2 library kit (Epicentre).

**ATAC-seq library construction and analysis.** E12.5 mouse hearts were dissected in ice cold optiMEM and cardiomyocytes were isolated using the Miltenyi Neonatal Mouse Cardiomyocyte Dissociation and Isolation Kits with the manufacturer's suggested protocol. The optimized-ATACSeq (omni-ATACSeq) libraries were generated from the isolated cardiomyocytes (two biological replicates; 26K cells each) using the author's recommended lysis steps and transposition protocol[52]. Libraries were sequenced using an Illumina NextSeq500 with single end 75 bp reads. Reads were mapped to the mm9 genome using Bowtie2[53]. Duplicates and reads mapping to blacklisted regions were removed using samtools[54]. Accessible regions were identified using MACS2[55] (macs2 callpeak -f BAM -g mm–keep-dup all–nomodel–nolambda–shift −100–extsize 200 -B -n).

**bioChIP-Seq library construction and analysis.** Libraries for bioChIP DNA and corresponding input samples were synthesized using KAPA Hyper Prep library kit (#KK8502). Each library was generated according to manufacturer's protocol. Cycle number for adapter-ligated libraries was determined by real-time PCR prior to amplification. Libraries were sequenced (single end, 75 bp) using an Illumina NextSeq500. BioChIP-Seq libraries were aligned against the mm9 genome using Bowtie2[53]. Duplicate reads and reads mapping to blacklist regions were removed. Peaks were called using MACS2 (macs2 callpeak -t chip-bl.bam -c input-bl.bam -f BAM -g mm -n chip -p 0.05 – verbose = 0) for ChIP samples with matched input samples using relaxed *P*-value (<0.05). Reproducible peaks were identified using IDR[56] at IDR_THRESHOLD = 0.05 between each set of replicate data files. All downstream analyses utilized a single TF bioChIP-Seq file generated from IDR and further processed so that each IDR peak was represented by the MACS2 summit ± 100 bp of the individual replicate with the greatest peak intensity.

H3K27ac-sequencing data for heart, forebrain, and lung of E11.5 and adult tissues were obtained from GSE52386, aligned to mm9, and peaks were called with MACS2 (macs2 callpeak -t H3K27ac.bam -c input.bam -f BAM -g mm -n chip -q 0.01 – verbose = 0) at FDR < 0.01. To define cardiac-enriched H3K27ac regions (cHRs), we created a region set containing the union of all peaks in these three tissues at each of the two developmental stages. We then calculated the normalized reads falling into these regions and used them to compute the cardiac H3K27ac score (CHS) = heart/max (liver, forebrain). Regions were ranked by CHS, and those in the top two quintiles were defined as cHRs.

**Bioinformatic analyses.** Deeptools[57] was used to analyze the correlation between bioChIP-seq samples and to generate aggregation plots.

To define regions "co-bound" by TFs, the single TF bioChIP-seq peak files were merged using the mergePeaks function of Homer[58], with parameter -d 300 (merge peaks whose centers are within 300 bp of one another).

Motif analysis was performed using Homer[58], which compares motif frequency in regions of interest compared to randomly selected sequences matched for GC content. Non-redundant significant motifs in the vertebrate Homer database were identified by motif clustering using STAMP[59] followed by manual inspection to select a motif representative of related motifs within each cluster. As an independent, complementary analysis, central enrichment of selected motifs within 1000 bp regions centered on peak summits was evaluated with Centrimo[28].

For composite motif analysis, we generated motif matrices for all four possible motif orientations with zero to eight intervening random bases. These matrices and Homer[58] were used to determine the enrichment of each composite motif within regions co-occupied by each pairwise combination of TFs. The variance of enrichment across the 32 possible composite motifs was determined by calculating the Fano factor (variance$^2$/mean).

Conservation of regions was analyzed using precalculated phastCons 30-way vertebrate scores[60].

Region overlaps were defined as regions that shared at least 1 base pair.

Intersection of the current data with previously reported murine ChIP-seq data (Supplementary Fig. 5) was performed using the R package Intervene[61].

Gene ontology analysis was performed using GREAT[62] using its default rule for associating regions to genes. In heatmaps summarizing GO analyses for several different conditions, we selected the 5 or 10 terms for each condition with the

highest statistical scores. The scores for the union of all terms over all the conditions is then represented as a heatmap of the negative $\log_{10}(P\text{-value})$.

To analyze the relationship between TF-bound regions and gene expression levels (Fig. 5a), we labeled each gene with a TF-binding number and a H3K27ac value (present or absent). The TF-binding number was the region within TSS ± 100 kb occupied by the greatest number of different TFs. If any of these regions with the greatest number of different TFs overlapped a H3K27ac region, then H3K27ac was assigned a value of 'present'.

Permutation analysis was performed using regioneR[63]. Regions were randomized 3000–10,000 times using the randomizeRegions module. Mappable regions of the genome were defined using 75mer alignment scores in mm9 (mm9wgEncodeCrgMapabilityAlign75mer.bigWig, downloaded from UCSC). Genome regions with mappability below 0.3 and length >500 bp were excluded. For computational efficiency, if a query region set contained >5000 regions, then it was divided into length/5000 sub-files containing 5000 randomly selected lines. The average result of permutation analysis of each of these sub-files was reported. Enrichment was defined as the observed overlap between a set of query regions and a set of target gene regions, compared to the overlap between random permutations of the query regions and the target gene regions. The target gene regions were defined as the gene TSS ± $d$, where $d$ varied from 5 to 240 kb as indicated.

**Enhancer prediction using machine learning.** Vista active enhancer database[41] was used as the gold standard for enhancer prediction. 1044 enhancers from mouse were scored as positive or negative for activity in heart or brain, as annotated in the Vista database. Intensities of factors (TF, number of TFs, H3K27ac, and ATAC) were used as features. Features were normalized using min–max normalization method. Two kinds of cross validation were used for evaluation.

1. All datasets were further divided into training (80%) and test (20%) sets. Xgboost package[64] was used to train an ensemble decision tree model using the following parameters: max_depth = 3; eta = 0.01; binary = logistic; eval_metric = logloss; subsample = 0.8). The model was boosted for 500 rounds, and the round (num_boost_round = 346) with the best evaluation score was used.
2. All datasets were validated using 100 permutations of five-fold cross-validation. Parameters were the same as above except that num_boost_round was set to 160 and eval_metric was set to auc.

**Reporting summary.** Further information on research design is available in the Nature Research Reporting Summary linked to this Article.

## Data availability

Sequencing data for this manuscript, summarized in Supplementary Tables 1 and 5, has been deposited into NCBI GEO database (GSE124008), which can be reviewed using this link: https://urldefense.proofpoint.com/v2/url?u=https-3A__www.ncbi.nlm.nih.gov_geo_query_acc.cgi-3Facc-3DGSE124008&d=DwIBAg&c=qS4goWBT7poplM69zy_3xhKwEW14JZMSdioCoppxeFU&r=CMWV1alzPmYOimiQcoBihLjlmPH2uRaUjet7j-VaCBttBhs6fqrkbUTGbYNA4QXXi&m=74CFVRnbOXksOek8m9wwHpMU3kfk0z-weIjNFlDtZQLw&s=Q7rzbA_8MH8x2VbEDfIp1RxWUjMCddq4CZMyYMOgeY0&e=. Data can also be accessed via the Cardiovascular Development Consortium server (https://b2b.hci.utah.edu/gnomex) (sign in as guest). Source data is available for figures presented in this manuscript and contains raw images for blots/gels and raw numbers used to generate graphs and plots. Raw source data for Figs. 1b, c, 2a, e, 3a, 7e and Supplementary Figs. 6b, 10h, 13b used aligned.bam files and/or BED files, which can be accessed at NCBI (#GSE124008) and Supplementary Data respectively.

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

## Acknowledgements

The authors thank the Boston Children's Gene Manipulation Core Facility for generation of epitope tagged knockin mice. This study was supported by the National Heart, Lung, and Blood Institute/Cardiovascular Development Consortium (UM1HL098166, U01HL098188, and U01HL131003). G.-C.Y. was supported by NIH R01HG009663. The content is solely the responsibility of the authors and does not necessarily represent the official views of the National Heart, Lung, and Blood Institute or the National Institutes of Health. Portions of this research were conducted on the O2 High Performance Compute Cluster, supported by the Research Computing Group, at Harvard Medical School. See http://rc.hms.harvard.edu for more information. pAV5S-F30-2xdBroccoli was a gift from Samie Jaffrey (Addgene plasmid #66845; http://n2t.net/addgene:66845; RRID:Addgene_66845).

## Author contributions

B.N.A. generated data. N.J.V. developed the MPRA method and performed the MPRA experiment. P.Z. contributed to ChIP-seq and performed RNA-seq from isolated cardiomyocytes. F.G., W.T.P., and B.N.A. analyzed the data. B.N.A. and W.T.P. wrote the manuscript with contributions from F.G. K.L. generated Mef2a[fb] and Mef2c[fb] mice. B. Zhang and B. Zhou generated Nkx2-5[fb] mice. L.W. and F.L.C. generated Tbx5[fb] mice. X.Z., I.S., R.D., and G.C.Y. contributed to data analysis. Q.M. performed echocardiography. T.W., J.L., K.D., and J.Z. contributed TEAD1 KO expression data.

## Competing interests

The authors declare no competing interests.
