## [Peer Review File · Nature Communications]

Reviewers' comments:

Reviewer #1 (Remarks to the Author):

This is an important contribution to the field of cardiac epigenetics and a well conducted investigation. The analyses are thorough—the comparisons with non-cardiac datasets are particularly useful to reveal general principles and the analysis of genetic relationships are an intuitive use of orthogonal data. I have the following suggestions for the authors:

I am curious about how the annotation was performed in figure 1d: are the 'distal/proximal' ratios in the middle panels based only on the peaks classified as 'TSS' in the bottom panel? Please explicitly clarify this in the text. Also, what is the difference between 'intergenic' and 'noncoding' in terms of how you classified these peaks?

In figure 1e: why do some of the columns not have assigned terms (i.e. color coded GO categories) beneath them...towards the right side of the panel?

Please reference the data to support this assertion: "Nuclear receptor motifs were not enriched amongst TF-bound regions in fetal heart, but showed enrichment in regions bound by several TFs in the adult heart."

The analysis of non-cardiac transcription factor motifs that were enriched in cardiac TF ChIP-seq peaks identified in this study is valuable and should be expanded to be more comprehensive. Which other TF motifs were enriched in your solo TF peaks and combined TF peaks, and were these TF motifs enriched in your peaks vs a random selection of comparable regions?

Please clarify which peaks were used for the clustering analyses in Figure 2. The text states that only distal peaks were used...does this mean all peaks that did not reside in TSS according to figure 1 annotation? Does this include all intergenic and intronic peaks? It would be very valuable for this analysis to be done on all peaks and for any differential behavior to be uncovered and separately reported. Example, is clustering more common in intergenic or intronic regions? How does this compare to TSS regions? Are different subsets of genes, based on annotation of the gene, regulated by nearby clustered TF binding events in introns versus intergenic regions?

Also related to Figure 2: it appears that the TF cobinding events were examined in ATAC peaks from fetal and adult heart separately. What if you examine a third group of ATAC peaks that change their accessibility between fetal and adult heart...is closing of chromatin associated with evacuation of TFs, and vice versa?

The circles indicating numbers of peaks should be proportional in figure 3b; currently, they are internally proportional but not between the analyses. Please indicate in the figure the numbers in each of the possible classifications revealed by these venn diagrams.

Reviewer #2 (Remarks to the Author):

The study by Akerberg et al reports generation of multiple lines of cardiac transcription factor knockin mice and bioinformatic analysis of factor binding in fetal and adult mice. Extensive data are included in the study generated using multiple seq approaches and informatic tools. The data are a rich source of TF binding and expression data that will be useful to the community. However, the reported data are not related to what is known of cardiac gene regulatory mechanisms and no new insights into cardiac biology are provided. In addition, there is little validation of identified enhancers or TF interactions beyond the in silico analyses. While the study includes a wealth of data to be mined, limited new insights into cardiac gene regulatory mechanisms or downstream biology in fetal or adult hearts are provided.

Specific Comments

1. While chromatin regions binding various cardiac TF were identified, the genes corresponding to these regions are not described in the study and the relevance to cardiac biology or gene regulatory mechanisms are not discussed.
2. Several cardiac enhancers with binding of multiple TFs have been reported over the years. Were any of these found in the reported data sets?
3. What is the biological significance of differences observed in the fetal and adult data sets? Likewise, the biological context of observed differences in the hybrid diversity panel and SNP

genotyping are not clear. Are there any novel findings there related to normal maturation of the heart or congenital heart disease mechanisms?

4. The functional validation of enhancers was not determined directly. Conservation and cardiac activity are described, but it is not clear what genes are affected or how they relate to cardiac development or function.

Reviewer #3 (Remarks to the Author):

Akerberg et al. define the occupancy of multiple cardiac transcription factors (TFs) in the fetal and adult heart and examine the relationship between TF occupancy, epigenetic marks and gene expression. Specifically, the paper presents bioChIP-seq using tagged knockin alleles of seven major cardiac TFs in fetal and adult heart; examines the relationship between TF and H3K27ac occupancy; examines the conservation of TF-occupied regions; examines whether stage-specific TF occupancy regulates stage-specific gene expression; examines the relationship between multi-TF bound regions and genes differentially expressed downstream of TF loss-of-function; and examines the ability of multi-TF co-occupancy to predict active heart enhancers.

Given that the majority of published datasets on the genomic localization of cardiac TFs have been performed in-vitro or using TF overexpression, the primary bioChIP-seq data for 7 cardiac TFs will represent an important addition to the field. From a resource perspective, this is a major contribution. However, the general claims concerning gene regulation are either similar to previous findings by this group or others, or are presented at a preliminary stage of analysis. Outside of the datasets as an important resource, the manuscript is not appropriate for a general journal such as Nature Communication in its present form.

Specific major concerns include:

- While a major advance, this “reference map” for cardiac TF occupancy is not comprehensive, in terms of TFs assessed, tissues assessed (ventricle only) and timepoints assessed. It is a major body of

work, but the limitations should be more overtly noted. Furthermore, this work extends the approach the authors have previously published (He et al Nature Communications 2014)

-There is little discussion of how concordant these data are with those existing data sets (e.g. Luna-Zurita et al Cell 2016). Given the importance of this manuscript as a resource, these comparisons should be considered.

- One of the central conclusions of the paper, that collaborative binding of cardiac TFs is functionally important for gene expression, has been previously considered by many labs including the authors of this manuscript.

- Certain features of multi-TF bound regions confound several of the analyses regarding other features of the regions (e.g. genomic distribution of multi-TF bound regions affects how likely they are to overlap ATAC/CHR regions).

- Both TF binding and H3K27Ac are quantitative traits and multiple reports support the model that they are quantitatively correlated with enhancer activity and target gene expression. However, the analysis included in this manuscript treats them both as binary. This is a significant oversimplification. For example, the major conclusion that many active enhancers bound by multiple TFs lack H3K27Ac signal may change if quantitative analysis is performed.

- The analysis of the relationship between TF-bound regions to gene expression is an important aspect of the paper but the conclusions are unconvincing. The functional genomics / expression results either don't support the hypothesis that multi-TF binding influences expression, or appear to show different sensitivity to TF number. Most of the expression change appears to occur by binding of 2-3 TFs. The eQTL data largely show that number of bound TFs does not affect likelihood of finding an eQTL within the TF-bound region, but the textual conclusions about these data suggest otherwise.

- The manuscript includes some analyses that show that multi-TF occupancy is a better predictor of functional importance than H3K27ac deposition, and some analyses that show the opposite - the authors should therefore avoid making general statements implying that multi-TF binding is a better feature for active enhancer prediction than H3K27ac deposition. Functional data would be necessary to make this point with clarity.

-The author focus mainly on the number of TFs occupying various regions but do not address the impact of specific TFs as carefully as they could. Correlation of gene expression changes in Gata4 and Tbx5 heterozygotes appears to suggest that gene expression changes are most clearly annotated to regions with 5+ TFs bound within the region. Regions with 5+ bound TFs only account for 3-5% of TF-occupied distal sites - this small number somewhat diminishes the importance of this set of regions as a reference map for future studies. Furthermore, it is challenging to interpret these gene expression changes without knowing how GATA4 or TBX5 occupancy are altered in the heterozygotes. More generally, the authors could more carefully consider the relationship of specific TFs in the clustered binding in relation to functional output.

Other specific concerns:

- The authors do not include comparison of bioChIP to Ab-based ChIP for any TFs. They claim that it is superior because of potential Ab “idiosyncrasies”, but need to show that it’s comparable in terms of binding regions and signal-to-noise ratio for the other TFs.

- Fig. 1b: The heat map is too large to not include sample names on both axes - it is hard to track samples across horizontally. The metric being correlated is unclear - number of reads within regions, area under curve, etc. Are the samples correlated in terms of signal strength at the same binding site, or in terms of binding site location? Also, could the stronger within-stage correlations be driven by the fact that a smaller number of regions were identified in the adult stage for each factor?

- Fig. 1c: What would a union set heatmap look like for all of the TF binding in the Fetal stage? It should have a larger number of shared sites than a union set heatmap for all of the TFs in the adult stage, according to the authors’ conclusions regarding correlation of stage-specific co-occupancy.

- Fig. 1e: How were the GO terms for enhancer sets, as opposed to gene expression, generated? Are the GO terms associated with the nearest gene to the TF? Or multiple neighboring genes? The text is unclear about this. The heatmap is also difficult to understand. What drives the emergence of the heart morphogenesis GO term in the adult heart for T and N?

Fig1. General. There is no consideration of regional specific expression of the analyzed TF and how that might impact their observations. Some have different expression in the LV vs. RV, some are myocardial specific whereas others are expressed in the endocardium and/or epicardium. None have consistent expression across the regions analyzed or lineages included.

- Fig. 2a: What do the authors suggest is the meaning of the 2nd high frequency peak in the distribution at a distance of approximately 10kb?

- Fig. 3b: need to include numbers or percentages in Venn diagrams - the difference is difficult to see by eye.

- Fig. 3c: How much of the increase in CHR overlap with regions with increasing number of TF binding sites is due to the fact that these sites are more likely to be proximal to a TSS (where there is often higher K27ac signal) rather than distal? Perhaps show K27ac enrichment for proximal and distal TF-bound regions separately. Does the percentage of TF-bound sites that overlap with CHR increase if you only look at those ChIP sites that also contain a binding motif for the given TF?

- Fig. 3e: Is the increased overlap of H3K27ac sites with ChIP sites that have more TF co-occupancy a result of the larger size of the regions that are more co-occupied?

- Fig. 3d: Is referred to in the text as “Fig. 4d”.

- Fig. 4a: Is there a general difference in genomic distribution / location for TF-bound sites vs H3K27ac sites that may be driving the difference in conservation observed?

- Fig. 4b: How much of the increase in conservation score in regions with increasing number of TF binding sites is due to the fact that these sites are more likely to be proximal to a TSS rather than distal? Perhaps show conservation scores for proximal and distal regions separately. Is there a general difference in genomic distribution / genomic annotation for TF-bound sites vs H3K27ac sites that may be driving the difference observed?

- Fig. 5a: Why are fetal no-CHR regions compared to the regions with 1 additional/less TF bound, but the fetal with-CHR regions are all compared to the regions bound by 1 TF?
- Fig. 5c: This figure and its explanatory text are unclear - how do the number of genes on the y-axis relate to the enrichment calculation? For example, why does the fetal-biased enrichment score increase with TSS distance, when the observed number of genes remains relatively similar for 1-4 TFs? Also, is the dotted line the randomly expected number? If so, wouldn't it be expected to increase with increasing distance from TSS as the search space grows?
- Fig. 5e: These data show no effect of multi-TF binding on eQTL enrichment, but the abstract includes eQTL enrichment as a metric favoring multi-TF binding. The data largely show that number of bound TFs does not affect likelihood of finding an eQTL within the TF-bound region, but the textual conclusions about these data suggest otherwise. We would suggest softening the conclusions. Are the TF regions used in this analysis specifically adult TF bound regions to match the adult eQTL data? What if you subset the eQTLs to those near genes with adult-biased expression?
- Fig. 6a/c: Are multi-TF bound sites enriched for DEGs to a higher extent than cHRs alone (not just multi-TF + cHR), thus making it a more reliable feature used to identify enhancers than cHR status?
- Fig. 7a: Is Mef2A occupancy within a VISTA region a more precise predictor of transgenic heart activity than 6 TF binding events? The very high recall seen for the individual features may be due to the fact that the VISTA enhancer regions are quite large.

We thank the reviewers for their constructive comments and appreciate their critical evaluation of our manuscript. Our revised manuscript addresses the reviewer's concerns with additional experiments, bioinformatic analyses and amended text. Among these new experiments are the first use of the massively parallel reporter assay in vivo within cardiomyocytes to measure cardiac enhancer activity of thousands of putative cardiac transcriptional regulatory regions. The manuscript has also been extensively rewritten to make it more focused and to fit within the Nature Comms. word limit.

Although all of the reviewers recognized the value of the TF map that we created as a resource for studies on heart development and disease, some reviewers questioned the advances and conceptual insights made in the study. We summarize here the major points of novelty. The manuscript was rewritten to better emphasize these points.

- TF occupancy reference map. We provide genome-wide TF occupancy maps that demonstrate the dramatic changes in TF binding patterns that occur between developmental stages.
- TF co-occupancy via indirect cooperative binding. The occupancy maps support the genome-wide collaborative binding of TFs to regulate cardiac gene expression. The data support indirect cooperative binding as the predominant mechanism that underlies collaborative cardiac TF binding.
- Highly conserved cardiac enhancers bound by multiple TFs. Our work illustrates that cardiac enhancers marked by TF occupancy are more highly conserved than those marked by H3K27ac. Whereas prior studies (Blow et al, 2010; Nord et al, 2013) based on H3K27ac or p300 occupancy found the cardiac enhancers are weakly conserved, our study suggests that cardiac enhancers defined by TF binding are well conserved.
- Strong activity of multiple-TF enhancers, irrespective of H3K27ac co-occupancy. In the revised manuscript, we provide direct evidence that regions co-occupied by multiple TFs, with or without H3K27ac co-occupancy, are potent cardiac enhancers. These measurements were made using a novel AAV-mediated massively reporter assay to measure enhancer activity in cardiomyocytes in vivo.
- Number of co-occupying TFs as an important feature for enhancer prediction. We used machine learning methods to compare the value of several different chromatin features to predict active cardiac enhancers. This analysis found that the most weighted feature was the number of co-occupying TFs, a feature that had not been previously considered in prior work in this area.
- TEAD1 as a central regulator of cardiomyocyte-specific biological processes. Using TEAD1 and NKX2-5 loss-of-function datasets, co-occupancy analysis, and motif analysis, we show that TEAD1 is a central regulator of cardiomyocyte-specific biological processes and metabolism. Given that TEAD1 is a major nuclear target of Hippo-YAP, this finding puts the Hippo-YAP pathway in a new context within the heart. We show that TEAD1 partners with NKX2-5, and that a preferred motif arrangement of TEAD1 and NKX2-5 is functionally significant for cardiac gene regulation.

Reviewer #1

“This is an important contribution to the field of cardiac epigenetics and a well conducted investigation. The analyses are thorough—the comparisons with non-cardiac datasets are particularly useful to reveal general principles and the analysis of genetic relationships are an intuitive use of orthogonal data. I have the following suggestions for the authors:”

We thank the reviewer for the positive comments and valuable feedback.

- 1) ***“I am curious about how the annotation was performed in figure 1d: are the ‘distal/proximal’ ratios in the middle panels based only on the peaks classified as ‘TSS’ in the bottom panel?”***

Please explicitly clarify this in the text. Also, what is the difference between ‘intergenic’ and ‘noncoding’ in terms of how you classified these peaks?’

In Figure 1d, the middle panel was based on the definition of “proximal” used throughout the manuscript, i.e. TSS +/- 2 kb. In the bottom panel, the classification of regions was performed by Homer: TSS means the region between TSS - 1 kb and TSS + 0.1 kb. The label “non-coding” referred to regions that express non-coding RNA, as annotated by Homer’s internal database. This was clarified with additional information in the figure legend, as well as relabeling “noncoding” as “ncRNA” in the revised figure.

2) “In figure 1e: why do some of the columns not have assigned terms (i.e. color coded GO categories) beneath them...towards the right side of the panel?”

The color coded GO categories were manually assigned into major functional groupings that served as summarized GO categories. The unlabeled columns did not fall into these major categories. We have revised Fig. 1e to ensure all columns are labeled. The full gene ontology list can be found in Supplementary Data 2.

3) “Please reference the data to support this assertion: ‘Nuclear receptor motifs were not enriched amongst TF-bound regions in fetal heart, but showed enrichment in regions bound by several TFs in the adult heart.’”

We have updated the text in the revised manuscript, to reference the supporting data (Supplementary Fig.6a) (page 4).

4) “The analysis of non-cardiac transcription factor motifs that were enriched in cardiac TF ChIP-seq peaks identified in this study is valuable and should be expanded to be more comprehensive. Which other TF motifs were enriched in your solo TF peaks and combined TF peaks, and were these TF motifs enriched in your peaks vs a random selection of comparable regions?”

The motif analysis displayed in Fig. 1f and Supplementary Fig. 6a measured enrichment compared to random background regions matched for GC content, as we clarified in the revised figure legends and methods. We used the entire vertebrate motif database from Homer, which contains 1297 motifs. We reduced the number of motifs by clustering them by motif similarity and selecting a motif to represent a cluster of related motifs. All significant non-redundant motifs are shown in Supplementary Fig. 6a. We did not show the analysis of all TF motifs for the combined TF regions because the resulting large table is similar to the aggregated result for individual factors making up the combined TF regions.

5) “Please clarify which peaks were used for the clustering analyses in Figure 2. The text states that only distal peaks were used...does this mean all peaks that did not reside in TSS according to figure 1 annotation? Does this include all intergenic and intronic peaks? It would be very valuable for this analysis to be done on all peaks and for any differential behavior to be uncovered and separately reported. Example, is clustering more common in intergenic or intronic regions? How does this compare to TSS regions? Are different subsets of genes, based on annotation of the gene, regulated by nearby clustered TF binding events in introns versus intergenic regions?”

In Figure 2, we used distal regions (greater than 2 kb from TSS) to avoid clustering that might be seen near the promoter. We have repeated the analysis of interpeak distance in the intronic and intergenic subsets as suggested by the reviewer (revised Supplementary Fig. 7a). The overall distribution and the distribution within intronic and intergenic regions was similar. The fraction of intronic or intergenic regions that were within the more closely spaced cluster was also similar. We revised the figure legend for Fig. 2 to clearly state that distal regions are plotted in the histogram.

- 6) ***“Also related to Figure 2: it appears that the TF cobinding events were examined in ATAC peaks from fetal and adult heart separately. What if you examine a third group of ATAC peaks that change their accessibility between fetal and adult heart...is closing of chromatin associated with evacuation of TFs, and vice versa?”***

We expanded this analysis to incorporate how changes in accessible chromatin may be associated with cobinding TFs in both fetal and adult bioChIP-seq datasets. We added revised Supplementary Fig. 8c to illustrate that stage-specific ATAC-seq regions are associated with stage-specific TF co-occupied regions. The main text has also been updated to include these results.

- 7) ***“The circles indicating numbers of peaks should be proportional in figure 3b; currently, they are internally proportional but not between the analyses. Please indicate in the figure the numbers in each of the possible classifications revealed by these venn diagrams.”***

We replotted the venn diagrams so that within a panel the areas are proportional to the set sizes, and each of the possible classifications are labeled with the number of regions.

Reviewer #2

- 1) ***“The study by Akerberg et al reports generation of multiple lines of cardiac transcription factor knockin mice and bioinformatic analysis of factor binding in fetal and adult mice. Extensive data are included in the study generated using multiple seq approaches and informatic tools. The data are a rich source of TF binding and expression data that will be useful to the community. However, the reported data are not related to what is known of cardiac gene regulatory mechanisms and no new insights into cardiac biology are provided. In addition, there is little validation of identified enhancers or TF interactions beyond the in silico analyses. While the study includes a wealth of data to be mined, limited new insights into cardiac gene regulatory mechanisms or downstream biology in fetal or adult hearts are provided.”***

The study provides new insights into cardiac gene regulatory mechanisms and downstream biology. We listed major novel insights in the introduction to this rebuttal.

In the revised manuscript, we include additional analyses and experiments to provide experimental validation of identified enhancers and selected TF interactions:

- An extensive literature search identified 52 regions that were previously shown to have cardiac enhancer activity, by transgenic analysis. Compared to random expectations, these validated cardiac enhancers were highly enriched for co-occupancy by multiple cardiac TFs (Supplementary Fig. 12c-d).
- We performed additional validation experiments using a novel reporter assay that measures enhancer activity in cardiomyocytes in vivo. We developed a massively parallel version of this assay and measured the activity of 2700 candidate enhancers in cardiomyocytes in vivo. This assay directly showed that TF occupancy, with or without H3K27ac, is sufficient for driving expression of mCherry in cardiomyocytes.
- We validated a subset of predicted TF interactions. In Supplementary Fig. 9, we show that TEAD1-MEF2C, TEAD1-NKX2-5, NKX2-5-MEF2C, and NKX2-5-TBX5 physically interact. In the revised manuscript, we further examined the impact of NKX2-5-TEAD1 interaction on differential gene expression downstream of TEAD1 knockout or NKX2-5 haploinsufficiency. (Fig. 7).

- 9) ***“While chromatin regions binding various cardiac TF were identified, the genes corresponding to these regions are not described in the study and the relevance to cardiac biology or gene regulatory mechanisms are not discussed.”***

The cardiac TFs regions are associated with the nearest neighboring gene in our revised Supplementary Data 1. The functional relevance of the TF-bound regions is the focus of revised Figures 4-7. We show (1) regions co-occupied by multiple TFs are sufficient for transcriptional enhancer activity in vivo (Fig. 4); and these regions are related to (2) strength of gene expression in fetal and adult heart and differential expression between fetal and adult heart (Fig. 6); (3) differentially expressed genes in NKX2-5 and TEAD1 heterozygous and homozygous knockout (Fig. 7).

10) ***“Several cardiac enhancers with binding of multiple TFs have been reported over the years. Were any of these found in the reported data sets?”***

Likely the reviewer is referring to cardiac promoters or regulatory regions that have been shown to be cooperatively regulated by different cardiac TFs using in vitro luciferase reporter assays. These studies typically use TF overexpression in heterologous systems, limiting extrapolation to the in vivo context and native expression levels. For example, we previously showed that in vitro luciferase reporter activity in cardiomyocytes vs. non-myocytes was not predictive of in vivo cardiac activity (He et al. 2011). The regulatory region that was studied most intensively in in vitro luciferase reporter assays for the collaborative action of cardiac TFs is the 0.7 kb *Nppa* promoter. In the revision, we've included *Nppa*-*Nppb* genome browser images to illustrate coordinated binding of multiple cardiac TFs and H3K27ac (Supplementary Fig. 5c-d). However, this region does not recapitulate the expression pattern of the native *Nppa* gene in space, time, or disease, as determined by in vivo transgenic assays (Horsthuis et al. 2008; Habets et al. 2002), reinforcing the need to interpret in vitro reporter assays in heterologous systems with caution.

To more systematically address the reviewer's question, we searched Pubmed using '(SRF mouse heart) OR (NKX2-5 mouse heart) OR (GATA4 mouse heart) OR (TBX5 mouse heart) OR (MEF2A mouse heart) OR (MEF2C mouse heart) OR (TEAD1 mouse heart)' to find examples in which in vivo regulation by these factors was established. Given the caveats of in vitro reporter assays, we focused on studies that analyzed enhancer activity in vivo in cardiomyocytes, i.e. transient transgenic enhancer assays, and excluded data that was limited to luciferase reporter assays. By these criteria, there are few examples that demonstrate in vivo activity of regions bound by multiple TFs. Out of 1522 articles retrieved by the search, we found 52 enhancers with demonstrated in vivo cardiac enhancer activity (this analysis excluded the VISTA enhancer database, which was analyzed separately). We determined the overlap of these 52 regions with cardiac TFs (Supplementary Table 2). Multi-TF regions were remarkably enriched in these 52 proven cardiac enhancers (30-200 fold enrichment over random expectation for regions bound by ≥ 5 TFs; $P < 0.0001$ by permutation testing; Supplementary Fig. 12c-d). This analysis provides additional evidence that many active cardiac enhancers are bound by multiple TFs.

11) ***“What is the biological significance of differences observed in the fetal and adult data sets? Likewise, the biological context of observed differences in the hybrid diversity panel and SNP genotyping are not clear. Are there any novel findings there related to normal maturation of the heart or congenital heart disease mechanisms?”***

In revised Fig. 6a-b, we examine the biological significance of the difference of TF binding observed between fetal and adult heart. We show that (1) strength of cardiac gene expression at each stage is related to TF occupancy pattern at the corresponding stage, and (2) genes with stage-specific expression are enriched near regions with stage specific multi-TF binding. To further illustrated this point in our revised manuscript, we have provided genome browser views of fetal- and adult-specific myosin heavy chains (*Myh7/Myh6*) and changing TF occupancy at these genes (Supplementary Fig. 4). These data suggest that dynamic changes in cardiac TF cobinding contribute to stage-specific gene expression.

12) ***“The functional validation of enhancers was not determined directly. Conservation and cardiac activity are described, but it is not clear what genes are affected or how they relate to cardiac development or function.”***

We used multiple analytical methods and available datasets to provide evidence that the regions occupied by multiple TFs have biological activity, including: (1) association with gene expression level; (2) association with stage-specific gene expression; (3) enrichment adjacent to genes differentially expressed by TF loss of function; (4) strong ability of TF binding number to predict cardiac enhancer activity.

However, to address the reviewer’s request for direct experimental evidence of enhancer activity by regions bound by multiple TFs, we performed additional analyses and experiments:

1. We analyzed previously identified cardiac enhancers and their overlap with our TF binding data in fetal heart. The 52 enhancers that we identified in our literature search were highly enriched for multi-TF binding (30-200 fold enrichment over random background for regions bound by ≥ 5 TFs; $P < 0.0001$; Supplementary Fig. 12d), providing direct experimental evidence that multi-TF regions have cardiac enhancer activity. Further reinforcing this observation, VISTA enhancer database regions with heart activity were highly enriched for overlap with multi-TF regions (4-10 fold enrichment over random expectation, $P < 0.0001$).
2. We performed in vivo enhancer activity assays by introducing enhancer-reporter constructs into cardiomyocytes in intact hearts using AAV9. In both individual assays and in a massively parallel reporter assay of 2,700 candidate regions, we found that regions bound by multiple (≥ 5) TFs were sufficient to promote transcription in vivo, with or without H3K27ac co-occupancy.

Reviewer #3

“Akerberg et al. define the occupancy of multiple cardiac transcription factors (TFs) in the fetal and adult heart and examine the relationship between TF occupancy, epigenetic marks and gene expression. Specifically, the paper presents bioChIP-seq using tagged knockin alleles of seven major cardiac TFs in fetal and adult heart; examines the relationship between TF and H3K27ac occupancy; examines the conservation of TF-occupied regions; examines whether stage-specific TF occupancy regulates stage-specific gene expression; examines the relationship between multi-TF bound regions and genes differentially expressed downstream of TF loss-of-function; and examines the ability of multi-TF co-occupancy to predict active heart enhancers.”

Given that the majority of published datasets on the genomic localization of cardiac TFs have been performed in-vitro or using TF overexpression, the primary bioChIP-seq data for 7 cardiac TFs will represent an important addition to the field. From a resource perspective, this is a major contribution.

Seven knockin mouse lines were generated for this study and used to produce 6-7 robust ChIP-seq datasets at two developmental time points. We thank the reviewer for recognizing the value of these mouse reagents (deposited at MMRRC) and our data on the binding of 7 TFs at endogenous expression levels in native heart. We searched the literature for genome-wide TF occupancy data in mouse cardiomyocytes or cardiomyocyte-like cells and found only 9 additional tracks of data on 3 of the seven factors, some without replicates. Most had significantly fewer peaks than identified using our bioChIP-seq approach.

- 1) ***“However, the general claims concerning gene regulation are either similar to previous findings by this group or others, or are presented at a preliminary stage of analysis. Outside of the datasets as an important resource, the manuscript is not appropriate for a general journal such as Nature Communication in its present form.”***

In addition to the valuable resource and mouse lines, the study provides new insights into cardiac gene regulatory mechanisms and downstream biology. The major points of novelty were summarized in the introduction to this rebuttal.

- 2) ***“While a major advance, this “reference map” for cardiac TF occupancy is not comprehensive, in terms of TFs assessed, tissues assessed (ventricle only) and timepoints assessed. It is a major body of work, but the limitations should be more overtly noted. Furthermore, this work extends the approach the authors have previously published (He et al Nature Communications 2014).”***

We agree that the TF occupancy map is not comprehensive in terms of cell type, developmental stage, or TFs studied and have indicated this in the revised discussion. The He et al. study mapped the binding of a single TF, GATA4, whereas this study extended that work to include six additional transcription factors. A major unexpected finding of the He et al. study was the large difference in TF binding sites between fetal and adult heart. By extending the analysis to several more TFs, which all displayed similar profound changes in TF occupancy, we can conclude that stage-specific TF occupancy is a general finding and not limited to only GATA4. Furthermore, we were able to analyze patterns of TF collaboration and binding, which led to the conclusion that indirect cooperative binding mechanisms are the predominant mechanism for TF collaborative binding. We determined the functional significance of multiple TF occupancy in the context of 1) putative H3K27ac-marked enhancers, 2) stage-specific gene expression and 3) TF loss-of-function. We have expanded on our analysis in this revised manuscript and updated the text to better highlight these points.

- 3) ***“There is little discussion of how concordant these data are with those existing data sets (e.g. Luna-Zurita et al Cell 2016). Given the importance of this manuscript as a resource, these comparisons should be considered.”***

We agree with the reviewer. In the revised manuscript, we expanded on our analyses by comparing previously reported TF binding data to the new data presented in this manuscript. We searched the literature for murine heart genome-wide occupancy data for the TFs that we studied. Our search of 1522 articles identified 6 genome-wide TF occupancy tracks for 3 TFs (NKX2-5, GATA4, and TBX3/5) in heart, 3 (NKX2-5, GATA4, and TBX5) in murine ESC-derived cardiomyocytes, and 5 in HL-1 cardiomyocyte-like cells (GATA4, MEF2A, NKX2-5, TBX5, SRF). We compared the overlap of these datasets to our data (Supplementary Fig. 5a-b). There was moderately good agreement between datasets, considering the differences in tissues studied and methodology. Our study identified more bound regions than prior studies, but most of the TF regions identified in the prior studies were contained within our data. Both Luna-Zurita et al. and Yen-Sin Ang et al. describe the Nppa-Nppb gene cluster and we have included IGV snapshots of that region to illustrate our TF binding data supports the observations found in those studies (Supplementary Fig. 5c-d).

- 4) ***“One of the central conclusions of the paper, that collaborative binding of cardiac TFs is functionally important for gene expression, has been previously considered by many labs including the authors of this manuscript.”***

Co-occupancy of multiple TFs at active enhancers was previously described by our lab (He et al. 2011) and others. Our prior study used HL-1 cardiomyocyte-like cells and TF overexpression. This study showed the value of the bioChIP-seq method, and the data from this study has been helpful for many subsequent studies. However, the cell type, in vitro context, and overexpression have led to questions about its biological relevance for heart development and disease. The data in the current manuscript reinforce the major finding of the prior study --, that multiple TF occupancy identifies functional enhancers. This study provides a more robust resource than our prior study -- it uses endogenous TF levels, in an in vivo context, and at two stages of heart development. Differences were observed between datasets

(Supplementary Fig. 5a-b), suggesting that the current in vivo data will be more applicable for in vivo studies of development and disease.

Furthermore, our analyses enhance and expand upon the prior conclusions. We provide multiple levels of functional evidence that the regulatory elements uncovered in this study have biological significance in vivo. We also show that a subset of these enhancers are functional without overlapping cardiac H3K27ac, and that their level of conservation is comparable to enhancers in other tissues. This is in contrast to the current paradigm, in which heart enhancers (identified by H3K27ac or P300 occupancy) are less conserved than enhancers in other tissues such as brain (Blow et al. 2010; Nord et al. 2013). Our data also suggest that the large majority of co-occupancy does not involve fixed TF-TF interactions at sites composed of composite motifs between two TFs (Jolma et al. 2015; Luna-Zurita et al. 2016). Rather, most cases of co-occupancy involve binding to regions that contain motifs for only a subset of the cobinding factors, and these motifs do not occur with defined spacing or orientation, which is consistent with facilitated binding models of TF-cobinding, e.g. cooperativity via joint histone eviction (Luebben, Sharma, and Nyborg 2010; Sharma and Nyborg 2008; Li et al. 2015). One exception to this was NKX2-5 and TBX5, which co-occupied regions with preferred motif arrangements that were consistent with those recently reported by Luna-Zurita et al. -- thereby extending the previous in vitro findings to an in vivo developmental context. A second exception was NKX2-5 and TEAD1, which also co-occupied regions with a preferred motif arrangement. Our analysis suggests that the preferred motif arrangement has functional significance. Finally, our study provides new context to Hippo-Yap signaling in the heart by showing that TEAD1, a major nuclear target of this signaling pathway, is an integral component of the cardiac transcriptional network and acts to regulate cardiomyocyte-specific biological functions.

Thus this paper provides an important resource and uses that resource to advance our understanding of the features that characterize TF cobinding and active enhancers and to elucidate aspects of cardiac transcriptional regulation. We rewrote sections of the manuscript to make these points more clear.

5) “Certain features of multi-TF bound regions confound several of the analyses regarding other features of the regions (e.g. genomic distribution of multi-TF bound regions affects how likely they are to overlap ATAC/CHR regions).”

ATAC and H3K27ac signals are both enriched at promoters, and TF binding was also enriched in these regions (promoter regions contained a higher density of TF occupancy when considering the percent of binding sites near the TSS and the percent of the genome near the TSS). For this reason, we excluded promoter regions from most of our analyses and focused on distal regions, which should be free from this confounding effect. We made this more clear in the revised manuscript by specifying in the figure legends whether all regions or distal regions were used for each analysis.

6) “Both TF binding and H3K27Ac are quantitative traits and multiple reports support the model that they are quantitatively correlated with enhancer activity and target gene expression. However, the analysis included in this manuscript treats them both as binary. This is a significant oversimplification. For example, the major conclusion that many active enhancers bound by multiple TFs lack H3K27Ac signal may change if quantitative analysis is performed.”

TF binding, H3K27ac, and ATAC are all quantitative traits, and this was incorporated into several analyses, such as the machine learning study (revised Fig. 5) and aspects of the other analyses (revised Fig. 2d,e; Fig. 3a,c-e; Supplementary Fig. 11d).

While addressing the reviewer’s critique, we re-examined our analyses of H3K27ac. We determined that it is confusing to consider regions as positive or negative for *cardiac* H3K27ac regions (those regions with H3K27ac signal preferentially in heart compared to forebrain or liver) because regions could have significant H3K27ac signal that was not cardiac specific. In the revised manuscript, the

primary focus is on heart H3K27ac and not its specificity for heart compared to forebrain or liver. We revised the text and figures accordingly.

For those portions of the analysis that used H3K27ac as a binary trait, defined by whether the level and distribution of H3K27ac signal was sufficient for MACS2 to detect a peak, we clarified the definition of the term “H3K27ac negative” in this study (Supplementary Fig. 11d): “At TF regions where H3K27ac was below the statistical threshold to call H3K27ac peaks, weak H3K27ac signal remained detectable above random background. This sub-threshold H3K27ac signal was on par with heart H3K27ac signal at regions with “forebrain-specific” or “liver-specific” H3K27ac occupancy. For clarity, hereafter we refer to these regions with subthreshold H3K27ac signal as “H3K27ac negative”.

7) “The analysis of the relationship between TF-bound regions to gene expression is an important aspect of the paper but the conclusions are unconvincing. The functional genomics / expression results either don’t support the hypothesis that multi-TF binding influences expression, or appear to show different sensitivity to TF number. Most of the expression change appears to occur by binding of 2-3 TFs. The eQTL data largely show that number of bound TFs does not affect likelihood of finding an eQTL within the TF-bound region, but the textual conclusions about these data suggest otherwise.”

We analyzed the relationship of the number of co-occupying TFs to TF binding, chromatin features, enhancer activity, and gene expression. Increasing number of co-occupying TFs was positively correlated to TF binding, chromatin accessibility (but not H3K27ac signal strength), conservation, and enrichment in/prediction of cardiac enhancers. The relationship of the number of co-occupying TFs to gene expression was more complex, with higher number being related (non-linearly) to gene expression level but without a consistent relationship to the likelihood that regions were enriched adjacent to genes with differential expression. This is not surprising, given that gene expression reflects the integrated activity of multiple regulatory regions rather than a single enhancer.

Given the large amount of data in this manuscript, space restrictions, and the relatively weak magnitude of the eQTL result, which was in part due to limited statistical power, we removed the eQTL analysis from the revised manuscript.

8) “The manuscript includes some analyses that show that multi-TF occupancy is a better predictor of functional importance than H3K27ac deposition, and some analyses that show the opposite - the authors should therefore avoid making general statements implying that multi-TF binding is a better feature for active enhancer prediction than H3K27ac deposition. Functional data would be necessary to make this point with clarity.”

We did not intend to claim that multi-TF occupancy is “better” than H3K27ac. The main point is that these are each predictive of a subset of enhancer regions, and that they are often non-overlapping. In the machine learning analysis, multi-TF occupancy was a stronger and non-redundant predictor of regions that are sufficient for cardiac enhancer activity. However, at the level of regulation of gene expression, multi-TF occupancy was not consistently associated with greater enrichment adjacent to differentially expressed genes, whereas H3K27ac occupancy was. We have updated the main results in the revised manuscript, and directly addressed it in the discussion: “Multi-TF regions were highly conserved and highly active in vivo in reporter assays, with H3K27ac co-occupancy having small to no overall effect on these functional measures. In contrast, cardiac H3K27ac regions without TF co-occupancy had weak conservation, as noted previously^{34,35}, and in the reporter assay had enhancer activity comparable to negative controls. However, at the level of gene expression we observed that H3K27ac co-occupied regions were enriched adjacent to DEGs, whereas there was no consistent pattern related to the number of co-occupying TFs. Likely this reflects the factors that determine the cis-regulatory activity of individual regions to compared the factors that determine overall gene expression, which reflects the complex integration of inputs from multiple cis-regulatory regions. Together these observations suggest that multi-

TF regions and H3K27ac regions are distinct classes of regulatory features that have different effects on transcriptional enhancer activity and on overall gene expression.”

- 9) ***“The author focus mainly on the number of TFs occupying various regions but do not address the impact of specific TFs as carefully as they could. Correlation of gene expression changes in Gata4 and Tbx5 heterozygotes appears to suggest that gene expression changes are most clearly annotated to regions with 5+ TFs bound within the region. Regions with 5+ bound TFs only account for 3-5% of TF-occupied distal sites - this small number somewhat diminishes the importance of this set of regions as a reference map for future studies. Furthermore, it is challenging to interpret these gene expression changes without knowing how GATA4 or TBX5 occupancy are altered in the heterozygotes. More generally, the authors could more carefully consider the relationship of specific TFs in the clustered binding in relation to functional output.”***

In the revised manuscript, the gene expression analyses focus on the effect of TEAD1, and its interaction with NKX2-5 (Fig. 7). NKX2-5 and TEAD1 frequently co-occupied the same regions (Fig. 2f), and we show that these factors physically interact (Supplementary Fig. 9). Given the interest in Hippo-YAP signaling, the relative lack of information on the role of TEAD1 (a major nuclear target of Hippo-YAP) in the heart, and our data showing that TEAD1 co-occupies regions in combination with cardiac TFs, we felt that focusing on TEAD1 would enhance the study.

We agree that it would be valuable to examine the effect of TF haploinsufficiency or knockout on the co-occupancy pattern of the other TFs. However, this is a challenging experiment using the bioChip system -- while the bioChIP system yields robust data, a weakness is that it is cumbersome to combine with different mouse genotypes. Robustly pursuing this line of investigation would require developing low input (e.g. single cell) bioChIP-seq methods so that these experiments could be performed in somatic knockouts.

- 10) ***“The authors do not include comparison of bioChIP to Ab-based ChIP for any TFs. They claim that it is superior because of potential Ab “idiosyncrasies”, but need to show that it’s comparable in terms of binding regions and signal-to-noise ratio for the other TFs.”***

Our lab has already validated and published the bioChIP-seq technique in three prior studies: (He et al. 2011, 2014; Zhou et al. 2017). These studies included comparison between antibody and biotin-streptavidin ChIP-seq. They showed that most Ab ChIP-seq regions are included in bioChIP-seq regions, but many bioChIP-seq regions are missed by Ab ChIP-seq. Interestingly, the He 2014 study showed that bioChIP-seq regions that were not detected by Ab ChIP-seq had a range of signal strength, i.e. it was not only the weak regions that were missed. We searched the literature for prior ChIP data from mouse heart or cardiomyocyte-like cells. We identified nine antibody-mediated ChIP-seq experiments, which yielded a smaller number of TF-bound regions that overlapped well with our chip-seq data, considering the differences in tissues/cells and methods between studies. Given that antibody reagents for many of the target TFs of this study are poor, it will not be productive to further validate the bioChIP-seq data by performing antibody ChIP-seq.

- 11) ***“Fig. 1b: The heat map is too large to not include sample names on both axes - it is hard to track samples across horizontally. The metric being correlated is unclear - number of reads within regions, area under curve, etc. Are the samples correlated in terms of signal strength at the same binding site, or in terms of binding site location? Also, could the stronger within-stage correlations be driven by the fact that a smaller number of regions were identified in the adult stage for each factor?”***

We modified the figure to include labels on the x-axis. We indicated the correlation metric in the revised figure legend. The number of regions did not drive the clustering since clustering used a union set of regions across all of the data tracks.

- 12) ***“Fig. 1c: What would a union set heatmap look like for all of the TF binding in the Fetal stage? It should have a larger number of shared sites than a union set heatmap for all of the TFs in the adult stage, according to the authors’ conclusions regarding correlation of stage-specific co-occupancy.”***

We have included a union set heatmap in the revised manuscript to illustrate this point (Fig. 2a).

- 13) ***“Fig. 1e: How were the GO terms for enhancer sets, as opposed to gene expression, generated? Are the GO terms associated with the nearest gene to the TF? Or multiple neighboring genes? The text is unclear about this. The heatmap is also difficult to understand. What drives the emergence of the heart morphogenesis GO term in the adult heart for T and N?”***

The top 1000 distal TF regions (sorted by binding strength) were associated with genes using the default method of GREAT, which generally associates the region with its two neighboring genes. The heatmap was created by making a union list of the 5 most significant GO biological process for each TF bioChIP-seq dataset. The significance score for the terms in the union list was displayed across all the TF bioChIP-seq datasets as a heatmap. This has been clarified in the revised figure legend and related methods.

- 14) ***“Fig 1. General. There is no consideration of regional specific expression of the analyzed TF and how that might impact their observations. Some have different expression in the LV vs. RV, some are myocardial specific whereas others are expressed in the endocardium and/or epicardium. None have consistent expression across the regions analyzed or lineages included.”***

All of the adult bioChIP-seq data in this manuscript relates exclusively to the lower (ventricular) portion of the adult heart, including both chambers. Fetal bioChIP-seq data was generated from isolated E12.5 hearts, which were further dissected to obtain the lower (ventricular) portion and to exclude the cardiac cushions and atria. In both stages, these tissues comprise cardiomyocytes and non-myocytes. This is a limitation of the current dataset, which we have stated in the revised discussion. We expanded the text in the methods section to describe what tissues were used.

- 15) ***“Fig. 2a: What do the authors suggest is the meaning of the 2nd high frequency peak in the distribution at a distance of approximately 10kb?”***

Second frequency peak represents the spacing between non-clustered binding sites. There were ~100,000 fetal and ~50,000 adult peaks with an interpeak distance > 1000 bp. Assuming that these are randomly distributed within a genome of 1E9 bp, the expected interpeak distance would be 10,000-20,000 bp, which is close to the center of the second frequency peak. We clarified this in the text on page 4: “We focused on regions distal (> 2 kb) to the TSS, to avoid potential effects of clustered TF binding near promoters. The inter-peak distance had a bimodal distribution (revised as Fig. 2b and Supplementary Fig. 7a). The local maximum around 10⁴ bp is close to the expected inter-peak distance for randomly distributed peaks, whereas with inter-peak distance less than 300 bp (dotted red line, Fig. 2b) indicates substantial clustering of cardiac TFs on chromatin.”

- 16) ***“Fig. 3b: need to include numbers or percentages in Venn diagrams - the difference is difficult to see by eye.”***

We replotted the Venn diagrams so that within a panel the ovals are proportional to the set sizes, and each of the possible classifications are labelled with the number of regions.

- 17) ***“Fig. 3c: How much of the increase in cHR overlap with regions with increasing number of TF binding sites is due to the fact that these sites are more likely to be proximal to a TSS (where there is often higher K27ac signal) rather than distal? Perhaps show K27ac enrichment for proximal and distal TF-bound regions separately. Does the percentage of TF-bound sites that overlap with cHR increase if you only look at those CHIP sites that also contain a binding motif for the given TF?”***

Our manuscript focuses on distal regulatory regions. All analyses looking at cHR region overlaps were limited to distal regions to eliminate any TF-promoter bias. We revised the text to make this more clear. We did not observe a relationship between presence/absence of a TF’s motif and the frequency of H3K27ac co-occupancy. Conversely, regions co-occupied by ≥ 5 TFs had on average 2-3 TF motifs, so that the number of TF binding motifs is unlikely to have driven the increased frequency of cHR or H3K27ac overlap.

- 18) ***“Fig. 3e: Is the increased overlap of H3K27ac sites with CHIP sites that have more TF co-occupancy a result of the larger size of the regions that are more co-occupied?”***

We observed that the average size for >5 TFs is between 590-702bp for adult and fetal multi-TF regions. To address this question, we set the size of all TF regions at 500 bp (cobound TF region center \pm 250 bp). We redid the H3K27ac - TF overlap analysis and observed that still over 50% of adult and fetal multi-TF regions (6 and 7 TFs respectively) overlap with cardiac-specific H3K27ac (Supplementary Fig. 11b).

- 19) ***“Fig. 3d: Is referred to in the text as “Fig. 4d”***

This mistake has been corrected in the revised text.

- 20) ***“Fig. 4a: Is there a general difference in genomic distribution / location for TF-bound sites vs H3K27ac sites that may be driving the difference in conservation observed?”***

The conservation analysis was limited to distal regions. This was clarified in the revised figure legend.

- 21) ***“Fig. 4b: How much of the increase in conservation score in regions with increasing number of TF binding sites is due to the fact that these sites are more likely to be proximal to a TSS rather than distal? Perhaps show conservation scores for proximal and distal regions separately. Is there a general difference in genomic distribution / genomic annotation for TF-bound sites vs H3K27ac sites that may be driving the difference observed?”***

All conservation analyses were done with distal regions, i.e. they did not include regions within $TSS \pm 2$ kb.

- 22) ***“Fig. 5a: Why are fetal no-cHR regions compared to the regions with 1 additional/less TF bound, but the fetal with-cHR regions are all compared to the regions bound by 1 TF?”***

We have updated this figure (revised Fig. 6a) to make the comparisons uniform. We have also updated the analysis to compare H3K27ac-marked regions in the heart, instead of only cardiac-specific H3K27ac (cHR).

- 23) ***“Fig. 5c: This figure and its explanatory text are unclear - how do the number of genes on the y-axis relate to the enrichment calculation? For example, why does the fetal-biased enrichment score increase with TSS distance, when the observed number of genes remains relatively similar for 1-4 TFs? Also, is the dotted line the randomly expected***

number? If so, wouldn't it be expected to increase with increasing distance from TSS as the search space grows?"

We apologize for the confusing labeling of the y-axis. The y-axis shows fold-enrichment of regions adjacent to either fetal-biased genes or adult-biased genes. Enrichment was calculated as the observed over expected value, where observed was the actual TF occupancy data, and random expectation was calculated by randomly permuting the TF occupancy data. We have improved the labeling of the figure (revised Fig. 6c) and its explanation in the figure legend.

24) ***"Fig. 5e: These data show no effect of multi-TF binding on eQTL enrichment, but the abstract includes eQTL enrichment as a metric favoring multi-TF binding. The data largely show that number of bound TFs does not affect likelihood of finding an eQTL within the TF-bound region, but the textual conclusions about these data suggest otherwise. We would suggest softening the conclusions. Are the TF regions used in this analysis specifically adult TF bound regions to match the adult eQTL data? What if you subset the eQTLs to those near genes with adult-biased expression?"***

We thank the reviewer for identifying the inaccuracy in the abstract. We have removed the cis-eQTL analyses from the revised manuscript.

25) ***"Fig. 6a/c: Are multi-TF bound sites enriched for DEGs to a higher extent than CHRs alone (not just multi-TF + CHR), thus making it a more reliable feature used to identify enhancers than CHR status?"***

In our revised manuscript, we have included H3K27ac regions without TFs (0TF+H3K27ac) in the DEG enrichment analyses. In general, in the revised study H3K27ac occupancy enriches regions adjacent to DEGs, whereas the number of co-occupying TFs is not a consistent determinant of the degree of enrichment. This differs from the analyses of the activity of individual enhancers, and likely reflects the difference between features that influence activity of individual regions compared to the integrated regulation of genes by multiple cis-regulatory regions.

26) ***"Fig. 7a: Is Mef2A occupancy within a VISTA region a more precise predictor of transgenic heart activity than 6 TF binding events? The very high recall seen for the individual features may be due to the fact that the VISTA enhancer regions are quite large."***

The reviewer is likely wondering why the recall and precision of individual factors is better than the recall and precision of the number of TFs in Fig. 7a (revised Fig. 5a). The reason is that TF number in 7a refers to regions with exactly that number of TFs. For example, Mef2a occupancy within a VISTA region is a more precise predictor of transgenic heart activity than occupancy by exactly 6 TFs, but 6-7 TFs would be more precise than Mef2a. It should also be clarified that the machine learning procedure considered occupancy of each TF as a quantitative trait (ChIP-seq signal strength) rather than a binary categorical trait. TF number was considered as an ordinal trait. Thus the parameters used for machine learning are slightly different than what is illustrated in Fig. 7a (revised Fig. 5a). To address the question of whether the relatively large size of VISTA regions increased the recall by individual features, we added as a negative control the precision/recall for prediction of brain enhancers.

References

Blow, M. J., D. J. McCulley, Z. Li, T. Zhang, J. A. Akiyama, A. Holt, I. Plajzer-Frick, et al. 2010. "ChIP-Seq Identification of Weakly Conserved Heart Enhancers." *Nature Genetics* 42 (9): 806–10.

Habets, P. E., A. F. Moorman, D. E. Clout, M. A. van Roon, M. Lingbeek, M. van Lohuizen, M. Campione, and V. M. Christoffels. 2002. "Cooperative Action of Tbx2 and Nkx2.5 Inhibits ANF Expression in the Atrioventricular Canal: Implications for Cardiac Chamber Formation." *Genes &*

Development 16 (10): 1234–46.

He, Aibin, Fei Gu, Yong Hu, Qing Ma, Lillian Yi Ye, Jennifer A. Akiyama, Axel Visel, Len A. Pennacchio, and William T. Pu. 2014. “Dynamic GATA4 Enhancers Shape the Chromatin Landscape Central to Heart Development and Disease.” *Nature Communications* 5: 4907.

He, Aibin, Sek Won Kong, Qing Ma, and William T. Pu. 2011. “Co-Occupancy by Multiple Cardiac Transcription Factors Identifies Transcriptional Enhancers Active in Heart.” *Proceedings of the National Academy of Sciences of the United States of America* 108 (14): 5632–37.

Horsthuis, T., A. C. Houweling, P. E. Habets, F. J. de Lange, H. El Azzouzi, D. E. Clout, A. F. Moorman, and V. M. Christoffels. 2008. “Distinct Regulation of Developmental and Heart Disease-Induced Atrial Natriuretic Factor Expression by Two Separate Distal Sequences.” *Circulation Research* 102 (7): 849–59.

Jolma, Arttu, Yimeng Yin, Kazuhiro R. Nitta, Kashyap Dave, Alexander Popov, Minna Taipale, Martin Enge, Teemu Kivioja, Ekaterina Morgunova, and Jussi Taipale. 2015. “DNA-Dependent Formation of Transcription Factor Pairs Alters Their Binding Specificity.” *Nature* 527 (7578): 384–88.

Li, Ming, Arjan Hada, Payel Sen, Lola Olufemi, Michael A. Hall, Benjamin Y. Smith, Scott Forth, et al. 2015. “Dynamic Regulation of Transcription Factors by Nucleosome Remodeling.” *eLife* 4 (June). <https://doi.org/10.7554/eLife.06249>.

Luebben, Whitney R., Neelam Sharma, and Jennifer K. Nyborg. 2010. “Nucleosome Eviction and Activated Transcription Require p300 Acetylation of Histone H3 Lysine 14.” *Proceedings of the National Academy of Sciences of the United States of America* 107 (45): 19254–59.

Luna-Zurita, Luis, Christian U. Stirnimann, Sebastian Glatt, Bogac L. Kaynak, Sean Thomas, Florence Baudin, Md Abul Hassan Samee, et al. 2016. “Complex Interdependence Regulates Heterotypic Transcription Factor Distribution and Coordinates Cardiogenesis.” *Cell* 164 (5): 999–1014.

Nord, Alex S., Matthew J. Blow, Catia Attanasio, Jennifer A. Akiyama, Amy Holt, Roya Hosseini, Sengthavy Phouanenvong, et al. 2013. “Rapid and Pervasive Changes in Genome-Wide Enhancer Usage during Mammalian Development.” *Cell* 155 (7): 1521–31.

Sharma, Neelam, and Jennifer K. Nyborg. 2008. “The Coactivators CBP/p300 and the Histone Chaperone NAP1 Promote Transcription-Independent Nucleosome Eviction at the HTLV-1 Promoter.” *Proceedings of the National Academy of Sciences of the United States of America* 105 (23): 7959–63.

Zhou, Pingzhu, Fei Gu, Lina Zhang, Brynn N. Akerberg, Qing Ma, Kai Li, Aibin He, et al. 2017. “Mapping Cell Type-Specific Transcriptional Enhancers Using High Affinity, Lineage-Specific Ep300 bioChIP-Seq.” *eLife* 6 (January). <https://doi.org/10.7554/eLife.22039>.

REVIEWERS' COMMENTS:

Reviewer #1 (Remarks to the Author):

The authors have addressed my concerns from the first review.

Reviewer #2 (Remarks to the Author):

My comments have been addressed and the MS is improved. The revised MS will serve as a useful resource for studies of transcriptional regulation of cardiac gene expression. The revised MS has increased biological impact with analysis of functional regulatory elements and emphasis on the central role of TEAD1.